# Interaction Hard Thresholding: Consistent Sparse Quadratic Regression in Sub-quadratic Time and Space

**Shuo Yang** *
Department of Computer Science
University of Texas at Austin
Austin, TX 78712
yangshuo_ut@utexas.edu

**Yanyao Shen** *
ECE Department
University of Texas at Austin
Austin, TX 78712
shenyanyao@utexas.edu

**Sujay Sanghavi**
ECE Department
University of Texas at Austin
Austin, TX 78712
sanghavi@mail.utexas.edu

## Abstract

Quadratic regression involves modeling the response as a (generalized) linear function of not only the features $\mathbf{x}^j$, but also of quadratic terms $\mathbf{x}^{j_1}\mathbf{x}^{j_2}$. The inclusion of such higher-order "interaction terms" in regression often provides an easy way to increase accuracy in already-high-dimensional problems. However, this explodes the problem dimension from linear $O(p)$ to quadratic $O(p^2)$, and it is common to look for sparse interactions (typically via heuristics).

In this paper we provide a new algorithm – Interaction Hard Thresholding (IntHT) – which is the first one to provably accurately solve this problem in *sub-quadratic* time and space. It is a variant of Iterative Hard Thresholding; one that uses the special quadratic structure to devise a new way to (approx.) extract the top elements of a $p^2$ size gradient in sub-$p^2$ time and space.

Our main result is to theoretically prove that, in spite of the many speedup-related approximations, IntHT linearly converges to a consistent estimate under standard high-dimensional sparse recovery assumptions. We also demonstrate its value via synthetic experiments.

Moreover, we numerically show that IntHT can be extended to higher-order regression problems, and also theoretically analyze an SVRG variant of IntHT.

## 1 Introduction

Simple linear regression aims to predict a response $y$ via a (possibly generalized) linear function $\boldsymbol{\theta}^\top \mathbf{x}$ of the feature vector $\mathbf{x}$. **Quadratic regression** aims to predict $y$ as a quadratic function $\mathbf{x}^\top \boldsymbol{\Theta} \mathbf{x}$ of the features $\mathbf{x}$

$$\begin{array}{cc} \text{Linear Model} & \text{Quadratic Model} \\ y \sim \boldsymbol{\theta}^\top \mathbf{x} & y \sim \mathbf{x}^\top \boldsymbol{\Theta} \mathbf{x} \end{array}$$

The inclusion of such higher-order *interaction terms* – in this case second-order terms of the form $\mathbf{x}^{j_1}\mathbf{x}^{j_2}$ – is common practice, and has been seen to provide much more accurate predictions in

---

several high-dimensional problem settings like recommendation systems, advertising, social network modeling and computational biology [23, 11, 3]. In this paper we consider quadratic regression with an additional (possibly non-linear) link function relating $y$ to $\mathbf{x}^\top \mathbf{\Theta} \mathbf{x}$.

One problem with explicitly adding quadratic interaction terms is that the dimension of the problem now goes from $p$ to $p^2$. In most cases, the quadratic problem is high-dimensional and will likely overfit the data; correspondingly, it is common to implicitly / explicitly impose low-dimensional structure on the $\mathbf{\Theta}$ – with sparsity of $\mathbf{\Theta}$ being a natural choice. A concrete example for sparse interaction would be the genome-wide association study, where for a given phenotype, the associated genetic variants are usually a sparse subset of all possible variants. Those genes usually interact with each other and leads to the given phenotype [15].

The **naive approach** to solving this problem involves recasting this as a big linear model that is now in $p^2$ dimensions, with the corresponding $p^2$ features being all pairs of the form $\mathbf{x}^{j_1} \mathbf{x}^{j_2}$. However, this approach takes $\Omega(p^2)$ time and space, since sparse linear regression cannot be done in time and space smaller than its dimension – which in this case is $p^2$ – even in cases where statistical properties like restricted strong convexity / incoherence etc. hold. Fundamentally, the problem lies in the fact that one needs to compute a gradient of the loss, and this is an $\Omega(p^2)$ operation.

**Our motivation:** Can we learn a sparse quadratic model with time and space complexity that is sub-quadratic? In particular, suppose we have data which is well modeled by a $\mathbf{\Theta}^*$ that is $K$-sparse, with $K$ being $\mathcal{O}(p^\gamma)$ and $\gamma < 1$. Statistically, this can be possibly recovered from $\mathcal{O}(K \log p)$ samples, each of which is $p$-dimensional. Thus we have a setting where the input is sub-quadratic with size $\mathcal{O}(Kp \log p)$, and the final output is sub-quadratic with size $\mathcal{O}(K)$. Our aim is to have an algorithm whose time and space complexity is also sub-quadratic for this case.

**In this paper**, we develop a new algorithm which has this desired sub-quadratic complexity, and subsequently theoretically establish that it consistently recovers a sparse $\mathbf{\Theta}^*$. We briefly overview our setting and results below.

## 1.1 Main Contributions

Given $n$ samples $\{(\mathbf{x}_i, y_i)\}_{i=1}^n$, we are interested in minimizing the following loss function corresponding to a quadratic model:

$$\textbf{(Quadratic Structure)} \qquad \min_{\mathbf{\Theta}: \|\mathbf{\Theta}\|_0 \leq K} \frac{1}{n} \sum_{i=0}^{n-1} f\left(\mathbf{x}_i^\top \mathbf{\Theta} \mathbf{x}_i, y_i\right) \quad := F_n\left(\mathbf{\Theta}\right) \qquad (1)$$

We develop a **new algorithm** – Interaction Hard Thresholding (IntHT), outlined in **Algorithm 1** – for this problem, and provide a **rigorous proof of consistency** for it under the standard settings (Restricted strong convexity and smoothness of the loss) for which consistency is established for sparse recovery problems. At a high level, it is based on the following **key ideas:**

**(1)** Because of the special quadratic structure, we show that the top $2k$ entries of the gradient can be found in sub-quadratic time and space, using ideas from hashing and coding. The subroutine in Algorithm 2 for doing this is based on the idea of [21] and **Theorem 1** characterizes its performance and approximation guarantee.

**(2)** We note a simple but key fact: in (stochastic) iterative hard thresholding, the new $k$-sparse $\mathbf{\Theta}_{t+1}$ that is produced has its support inside the union of two sets of size $k$ and $2k$: the support of the previous $\mathbf{\Theta}_t$, and the top-$2k$ elements of the gradient.

**(3)** While we do not find the precise top-$2k$ elements of the gradient, we do find an approximation. Using a new theoretical analysis, we show that this approximate-top-$2k$ is still sufficient to establish linear convergence to a consistent solution. This is our main result, described in **Theorem 4**.

**(4)** As an extension, we show that our algorithm also works with popular SGD variants like SVRG (**Algorithm 4 in Appendix B**), with provable linear convergence and consistency **in Appendix C**. We also demonstrate the extension of our algorithm to estimate higher order interaction terms with a numerical experiment **in Section 5** .

**Notation** We use $[n]$ to represent the set $\{0, \cdots, n-1\}$. We use $f_{\mathcal{B}}\left(\mathbf{\Theta}\right)$ to denote the average loss on batch $\mathcal{B}$, where $\mathcal{B}$ is a subset of $[n]$ with batch size $m$. We define $\langle \mathbf{A}, \mathbf{B} \rangle = \mathrm{tr}\left(\mathbf{A}^\top \mathbf{B}\right)$, and $\mathrm{supp}(\mathbf{A})$

to be the index set of $\mathbf{A}$ with non-zero entries. We let $\mathcal{P}_S$ to be the projection operator onto the index set $S$. We use standard Big-$\mathcal{O}$ notation for time/space complexity analysis, and Big-$\widetilde{\mathcal{O}}$ notation which ignores log factors.

## 2    Related Work

**Learning with high-order interactions** Regression with interaction terms has been studied in the statistics community. However, many existing results consider under the assumption of strong/weak hierarchical (SH/WH) structure: the coefficient of the interaction term $\mathbf{x}^{j_1}\mathbf{x}^{j_2}$ is non-zero only when both coefficients of $\mathbf{x}^{j_1}$ and $\mathbf{x}^{j_2}$ are (or at least one of them is) non-zero. Greedy heuristics [32, 11] and regularization based methods [7, 3, 16, 25, 10] are proposed accordingly. However, they could potentially miss important signals that only contains the effect of interactions. Furthermore, several of these methods also suffer from scaling problems due to the quadratic scaling of the parameter size. There are also results considering the more general tensor regression, see, e.g., [34, 9], among many others. However, neither do these results focus on solutions with efficient memory usage and time complexity, which may become a potential issue when the dimension scales up. From a combinatorial perspective, [18, 13] learns sparse polynomial in Boolean domain using quite different approaches.

**Sparse recovery, IHT and stochastic-IHT** IHT [4] is one type of sparse recovery algorithms that is proved to be effective for M-estimation [12] under the regular RSC/RSM assumptions. [20] proposes and analyzes a stochastic version of IHT. [14, 26] further consider variance reduced acceleration algorithm under this high dimensional setting, [35] studies IHT in high dimensional setting with nonlinear measurement. Notice that IHT, if used for our quadratic problem, still suffers from quadratic space, similar to other techniques, e.g., the Lasso, basis pursuit, least angle regression [29, 6, 8]. On the other hand, [19] recently considers a variant of IHT, where for each sample, only a random subset of features is observed. This makes each update cheap, but their sample size has linear dependence on the ambient dimension, which is again quadratic. Apart from that, [20, 17] also show that IHT can potentially tolerate a small amount of error per iteration .

**Maximum inner product search** One key technique of our method is extracting the top elements (by absolute value) of gradient matrix, which can be expressed as the inner product of two matrices. This can be formulated as finding Maximum Inner Product (MIP) from two sets of vectors. In practice, algorithms specifically designed for MIP are proposed based on locality sensitive hashing [27], and many other greedy type algorithms [2, 33]. But they either can't fit into the regression setting, or suffers from quadratic complexity. In theory, MIP is treated as a fundamental problem in the recent development of complexity theory [1, 31]. [1, 5] shows the hardness of MIP, even for Boolean vectors input. While in general hard, there are data dependent approximation guarantees, using the compressed matrix multiplication method [21], which inspired our work.

**Others** The quadratic problem we study also share similarities with several other problem settings, including factorization machine [23] and kernel learning [24, 22]. Different from factorization machine, we do not require the input data to be sparse. While the factorization machine tries to learn a low rank representation, we are interested in learning a sparse representation. Compared to kernel learning, especially the quadratic / polynomial kernels, our task is to do feature selection and identify the correct interactions.

## 3    Interaction Hard Thresholding

We now describe the main ideas motivating our approach, and then formally describe the algorithm.

**Naively recasting as a linear model has $p^2$ time and space complexity:** As a first step to our method, let us see what happens with the simplest approach. Specifically, as noted before, problem (1) can be recast as one of finding a sparse (generalized) linear model in the $p^2$ size variable $\mathbf{\Theta}$:

$$\text{(Recasting as linear model)} \qquad \min_{\mathbf{\Theta}:\|\mathbf{\Theta}\|_0 \leq K} \frac{1}{n}\sum_{i=0}^{n-1} f\left(\langle \mathbf{X}_i, \mathbf{\Theta}\rangle, y_i\right)$$

where matrix $\mathbf{X}_i := \mathbf{x}_i\mathbf{x}_i^\top$. Iterative hard thresholding (IHT) [4] is a state-of-the-art method (both in terms of speed and statistical accuracy) for such sparse (generalized) linear problems. This involves

---

**Algorithm 1** INTERACTION HARD THRESHOLDING (INTHT)

1: **Input:** Dataset $\{\mathbf{x}_i, y_i\}_{i=1}^n$, dimension $p$
2: **Parameters:** Step size $\eta$, estimation sparsity $k$, batch size $m$, round number $T$
3: **Output:** The parameter estimation $\widehat{\Theta}$
4: Initialize $\Theta^0$ as a $p \times p$ zero matrix.
5: **for** $t = 0$ **to** $T - 1$ **do**
6:     Draw a subset of indices $\mathcal{B}_t$ from $[n]$ randomly.
7:     Calculate the residual $u_i = u(\Theta^t, \mathbf{x}_i, y_i)$ based on eq. (2), for every $i \in \mathcal{B}_t$.
8:     Set $\mathbf{A}_t \in \mathbb{R}^{p \times m}$, where each column of $\mathbf{A}_t$ is $u_i \mathbf{x}_i$, $i \in \mathcal{B}_t$.
9:     Set $\mathbf{B}_t \in \mathbb{R}^{p \times m}$, where each column of $\mathbf{B}_t$ is $\mathbf{x}_i$, $i \in \mathcal{B}_t$. (where $\frac{\mathbf{A}_t \mathbf{B}_t^\top}{m}$ gives the gradient)
10:    Compute $\widetilde{S}_t = \text{ATEE}(\mathbf{A}_t, \mathbf{B}_t, 2k)$.    —/* *approximate top elements extraction* */—
11:    Set $S_t = \widetilde{S}_t \cup \text{supp}(\Theta^t)$.    —/* *inaccurate hard thresholding update* */—
12:    Compute $\mathcal{P}_{S_t}(\mathbf{G}^t) \leftarrow$ the gradient value $\mathbf{G}^t = \frac{1}{m} \sum_{i \in \mathcal{B}^t} u_i \mathbf{x}_i \mathbf{x}_i^\top$ only calculated on $S_t$.
13:    Update $\Theta^{t+1} = \mathcal{H}_k \left( \Theta^t - \eta \mathcal{P}_{S_t}(\mathbf{G}^t) \right)$.
14: **Return:** $\widehat{\Theta} = \Theta^T$

---

**Algorithm 2** APPROXIMATED TOP ELEMENTS EXTRACTION (ATEE)

1: **Input:** Matrix $\mathbf{A}$, matrix $\mathbf{B}$, top selection size $k$
2: **Parameters:** Output set size upper bound $b$, repetition number $d$, significant level $\Delta$
3: **Expected Output:** Set $\Lambda$: the top-$k$ elements in $\mathbf{AB}^\top$ with absolute value greater than $\Delta$
4: **Output:** Set $\widetilde{\Lambda}$ of indices, with size at most $b$ (approximately contains $\Lambda$)

---

5: **Short Description:** This algorithm is adopted directly from [21]. It follows from the matrix compressed product via FFT (see section 2.2 of [21]) and sub-linear result extraction by error-correcting code (see section 4 of [21]), which drastically reduces the complexity. The whole process is repeated for $d$ times to boost the success probability. The notation here matches [21] exactly, except that we use $p$ for dimension while $n$ is used in [21] instead.
6: Intuitively, the algorithm will put all the elements of $\mathbf{AB}^\top$ into b different "basket"s, with each of the elements assigned a positive or negative sign. It then selects the "basket" whose magnitude is greater than $\Delta$. Further, one large element is recovered from each of the selected baskets.

---

the following update rule

$$\text{(standard IHT)} \qquad \Theta^{t+1} = \mathcal{H}_k \left( \Theta^t - \eta \nabla F_n(\Theta^t) \right)$$

where $F_n(\cdot)$ is the average loss defined in (1), and $\mathcal{H}_k(\cdot)$ is the hard-thresholding operator that chooses the largest $k$ elements (in terms of absolute value) of the matrix given to it, and sets the rest to 0. Here, $k$ is the estimation sparsity parameter. In this update equation, the current iterate $\Theta^t$ has $k$ non-zero elements and so can be stored efficiently. But the gradient $\nabla F_n(\Theta^t)$ is $p^2$ dimensional; this causes IHT to have $\Omega(p^2)$ complexity. This issue remains even if the gradient is replaced by a stochastic gradient that uses fewer samples, since even in a stochastic gradient the number of variables remains $p^2$.

**A key observation:** We only need to know the top-$2k$ elements of this gradient $\nabla F_n(\Theta^t)$, because of the following simple fact: if $\mathbf{A}$ is a $k$-sparse matrix, and $\mathbf{B}$ is any matrix, then

$$\text{supp}(\mathcal{H}_k(\mathbf{A} + \mathbf{B})) \subset \text{supp}(\mathbf{A}) \cup \text{supp}(\mathcal{H}_{2k}(\mathbf{B})).$$

That is, the support of the top $k$ elements of the sum $\mathbf{A} + \mathbf{B}$ is inside the union of the support of $\mathbf{A}$, and the top-$2k$ elements of $\mathbf{B}$. The size of this union set is at most $3k$.

Thus, in the context of standard IHT, we do not really need to know the full (stochastic) gradient $\nabla F_n(\Theta^t)$; instead we only need to know (a) the values and locations of its top-$2k$ elements, and (b) evaluate at most $k$ extra elements of it – those corresponding to the support of the current $\Theta^t$.

The **key idea** of our method is to exploit the special structure of the quadratic model to find the top-$2k$ elements of the batch gradient $\nabla f_{\mathcal{B}}$ in sub-quadratic time. Specifically, $\nabla f_{\mathcal{B}}$ has the following form:

$$\nabla f_{\mathcal{B}}(\Theta) \triangleq \frac{1}{m} \sum_{i \in \mathcal{B}} \nabla f \left( \mathbf{x}_i^\top \Theta \mathbf{x}_i, y_i \right) = \frac{1}{m} \sum_{i \in \mathcal{B}} u(\Theta, \mathbf{x}_i, y_i) \mathbf{x}_i \mathbf{x}_i^\top, \qquad (2)$$

where $u(\mathbf{\Theta}, \mathbf{x}_i, y_i)$ is a scalar related to the residual and the derivative of link function , and $\mathcal{B}$ represents the mini-batch where $\mathcal{B} \subset [n], |\mathcal{B}| = m$. This allows us to approximately find the top-$2k$ elements of the $p^2$-dimensional stochastic gradient in $\widetilde{\mathcal{O}}(k(p+k))$ time and space, which is sub-quadratic when $k$ is $\mathcal{O}(p^\gamma)$ for $\gamma < 1$.

Our algorithm is formally described in Algorithm 1. We use Approximate Top Elements Extraction (ATEE) to approximately find the top-$2k$ elements of the gradient, which is briefly summarized in Algorithm 2, based on the idea of Pagh [21]. The full algorithm is re-organized and provided in Appendix A for completeness. Our method, Interaction Hard Thresholding (IntHT) builds on IHT, but needs a substantially new analysis for proof of consistency. The subsequent section goes into the details of its analysis.

## 4  Theoretical Guarantees

In this section, we establish the consistency of Interaction Hard Thresholding, in the standard setting where sparse recovery is established.

Specifically, we establish convergence results under deterministic assumptions on the data and function, including restricted strong convexity (RSC) and smoothness (RSM). Then, we analyze the sample complexity when features are generated from sub-gaussian distribution in the quadratic regression setting, in order to have well-controlled RSC and RSM parameters. The analysis of required sample complexity yields an overall complexity that is sub-quadratic in time and space.

### 4.1  Preliminaries

We first describe the standard deterministic setting in which sparse recovery is typically analyzed. Specifically, the samples $(\mathbf{x}_i, y_i)$ are fixed and known. Our first assumption defines how our intended recovery target $\mathbf{\Theta}^\star$ relates to the resulting loss function $F_n(\cdot)$.

**Assumption 1** (Standard identifiability assumption). *There exists a $\mathbf{\Theta}^\star$ which is $K$-sparse such that the following holds: given any batch $\mathcal{B} \subset [n]$ of $m$ samples, the norm of batch gradient at $\mathbf{\Theta}^\star$ is bounded by constant $G$. That is, $\|\nabla f_{\mathcal{B}}(\mathbf{\Theta}^\star)\|_F \leq G$, and $\|\mathbf{\Theta}^\star\|_\infty \leq \omega$.*

In words, this says the the gradient at $\mathbf{\Theta}^\star$ is small. In a noiseless setting where data is generated from $\mathbf{\Theta}^\star$, e.g. when $y_i = \mathbf{x}_i^\top \mathbf{\Theta}^\star \mathbf{x}_i$, this gradient is 0; i.e. the above is satisfied with $G = 0$, and $\mathbf{\Theta}^\star$ would be the exact sparse optimum of $F_n(\cdot)$. The above assumption generalizes this notion to noisy and non-linear cases, relating our recovery target $\mathbf{\Theta}^\star$ to the loss function. This is a standard setup assumption in sparse recovery.

Now that we have specified what $\mathbf{\Theta}^\star$ is and why it is special, we specify the properties the loss function needs to satisfy. These are again standard in the sparse recovery literature [20, 26, 14].

**Assumption 2** (Standard landscape properties of the loss). *For any pair $\mathbf{\Theta}_1, \mathbf{\Theta}_2$ and $s \leq p^2$ such that $|\text{supp}(\mathbf{\Theta}_1 - \mathbf{\Theta}_2)| \leq s$*

- *The overall loss $F_n$ satisfies $\alpha_s$-Restricted Strong Convexity (RSC):*

$$F_n(\mathbf{\Theta}_1) - F_n(\mathbf{\Theta}_2) \geq \langle \mathbf{\Theta}_1 - \mathbf{\Theta}_2, \nabla_{\mathbf{\Theta}} F_n(\mathbf{\Theta}_2) \rangle + \frac{\alpha_s}{2} \|\mathbf{\Theta}_1 - \mathbf{\Theta}_2\|_F^2$$

- *The mini-batch loss $f_{\mathcal{B}}$ satisfies $L_s$-Restricted Strong Smoothness (RSM):*

$$\|\nabla f_{\mathcal{B}}(\mathbf{\Theta}_1) - \nabla f_{\mathcal{B}}(\mathbf{\Theta}_2)\|_F \leq L_s \|\mathbf{\Theta}_1 - \mathbf{\Theta}_2\|_F, \ \forall \mathcal{B} \subset [n], \ |\mathcal{B}| = m$$

- *$f_{\mathcal{B}}$ satisfies Restricted Convexity (RC) (but not strong):*

$$f_{\mathcal{B}}(\mathbf{\Theta}_1) - f_{\mathcal{B}}(\mathbf{\Theta}_2) - \langle \nabla f_{\mathcal{B}}(\mathbf{\Theta}_2), \mathbf{\Theta}_1 - \mathbf{\Theta}_2 \rangle \geq 0, \ \forall \mathcal{B} \subset [n], \ |\mathcal{B}| = m, \ s = 3k + K$$

**Note:** While our assumptions are standard, our result does not follow immediately from existing analyses – because we cannot find the exact top elements of the gradient. We need to do a new analysis to show that even with our approximate top element extraction, linear convergence to $\mathbf{\Theta}^\star$ still holds.

## 4.2 Main Results

Here we proceed to establish the sub-quadratic complexity and consistency of IntHT for parameter estimation. Theorem 1 presents the analysis of ATEE. It provides the computation complexity analysis, as well as the statistical guarantee of support recovery. Based on this, we show the per round convergence property of Algorithm 1 in Theorem 3. We then establish our main statistical result, the linear convergence of Algorithm 1 in Theorem 4.

Next, we discuss the batch size that guarantees support recovery in Theorem 5, focusing on the quadratic regression setting, i.e. the model is linear in both interaction terms and linear terms. Combining all the established results, the sub-quadratic complexity is established in Corollary 6. All the proofs in this subsection can be found in Appendix E.

**Analysis of ATEE** Consider ATEE with parameters set to be $b, d, \Delta$. Recall this means that ATEE returns an index set $(\widetilde{\Lambda})$ of size at most $b$, which is expected to contain the desired index set $(\Lambda)$. Note that the desired index set $(\Lambda)$ is composed by the top-$2k$ elements of gradient $\nabla f_{\mathcal{B}}(\Theta)$ whose absolute value is greater than $\Delta$. Suppose now the current estimate is $\Theta$, and $\mathcal{B}$ is the batch. The following theorem establishes when this output set $(\widetilde{\Lambda})$ captures the top elements of the gradient.

**Theorem 1** (Recovering top-$2k$ elements of the gradient, modified from [21])**.** *With the setting above, if we choose $b, d, \Delta$ so that $b\Delta^2 \geq 432 \|\nabla f_{\mathcal{B}}(\Theta)\|_F^2$ and $d \geq 48 \log 2ck$, then the index set $(\widetilde{\Lambda})$ returned by ATEE contains the desired index set $(\Lambda)$ with probability at least $1 - 1/c$.*

*Also in this case the time complexity of ATEE is $\widetilde{\mathcal{O}}(m(p+b))$, and space complexity is $\widetilde{\mathcal{O}}(m(p+b))$.*

Theorem 1 requires that parameter $b, \Delta$ are set to satisfy $b\Delta^2 \geq 432 \|\nabla f_{\mathcal{B}}(\Theta)\|_F^2$. Note that $\Delta$ controls the minimum magnitude of top-$k$ element we can found. To avoid getting trivial extraction result, we need to set $\Delta$ as a constant that doesn't scale with $p$. In order to control the scale of $\Delta$ and $b$, to get consistent estimation and to achieve sub-quadratic complexity, we need to upper bound $\|\nabla f_{\mathcal{B}}(\Theta)\|_F^2$. This is the *compressibility estimation* problem that was left open in [21]. In our case, the batch gradient norm can be controlled by the RSM property. More formally, we have

**Lemma 2** (Frobenius norm bound of gradient)**.** *The Frobenius norm of batch gradient at arbitrary $k$-sparse $\Theta$, with $\|\Theta\|_\infty \leq \omega$, can be bounded as $\|\nabla f_{\mathcal{B}}(\Theta)\|_F \leq 2L_{2k}\sqrt{k}\omega + G$, where $G$ is the uniform bound on $\|\nabla f_{\mathcal{B}}(\Theta^\star)\|_F$ over all batches $\mathcal{B}$ and $\omega$ bounds $\|\Theta^\star\|_\infty$ (see Assumption 1).*

Lemma 2 directly implies that Theorem 1 could allow $b$ scale linearly with $k$ while keep $\Delta$ as a constant[2]. This is the key ingredient to achieve sub-quadratic complexity and consistent estimation. We postpone the discussion for complexity to later paragraph, and proceed to finish the statistical analysis of gradient descent.

**Convergence of IntHT:** Consider IntHT with parameter set to be $\eta, k$. For the purpose of analysis, we keep the definition of $\Lambda$ and $\widetilde{\Lambda}$ from the analysis of ATEE and further define $k_\Delta$ to be the number of top-$2k$ elements whose magnitude is below $\Delta$. Recall that $K$ is the sparsity of $\Theta^\star$, define $\nu = 1 + \left(\rho + \sqrt{(4+\rho)\rho}\right)/2, \rho = K/k$, where $\nu$ measures the error induced by exact IHT (see Lemma 9 for detail). Denote $B_t = \{\mathcal{B}_0, \mathcal{B}_1, ..., \mathcal{B}_t\}$. We have

**Theorem 3** (Per-round convergence of IntHT)**.** *Following the above notations, the per-round convergence of Algorithm 1 satisfies the following:*

- *If ATEE succeeds, i.e., $\Lambda \subseteq \widetilde{\Lambda}$, then*

$$\mathbb{E}_{B_t}\left[\left\|\Theta^t - \Theta^\star\right\|_F^2\right] \leq \kappa_1 \mathbb{E}_{B_{t-1}}\left[\left\|\Theta^{t-1} - \Theta^\star\right\|_F^2\right] + \sigma_{GD}^2 + \sigma_{\Delta|GD}^2,$$

  *where $\kappa_1 = \nu\left(1 - 2\eta\alpha_{2k} + 2\eta^2 L_{2k}^2\right), \sigma_{\Delta|GD}^2 = 4\sqrt{k_\Delta}\eta\sqrt{k}\omega\Delta + 2k_\Delta\eta^2\Delta^2$, and*

$$\sigma_{GD}^2 = \max_{|\Omega| \leq 2k+K}\left[4\nu\eta\sqrt{k}\omega \|\mathcal{P}_\Omega\left(\nabla F\left(\Theta^\star\right)\right)\|_F + 2\nu\eta^2 \mathbb{E}_{\mathcal{B}_t}\left[\|\mathcal{P}_\Omega\left(\nabla f_{\mathcal{B}_t}\left(\Theta^\star\right)\right)\|_F^2\right]\right].$$

- *If ATEE fails, i.e., $\Lambda \not\subset \widetilde{\Lambda}$, then,*

$$\mathbb{E}_{B_t}\left[\left\|\Theta^t - \Theta^\star\right\|_F^2\right] \leq \kappa_2 \mathbb{E}_{B_{t-1}}\left[\left\|\Theta^{t-1} - \Theta^\star\right\|_F^2\right] + \sigma_{GD}^2 + \sigma_{Fail|GD}^2,$$

$$\text{where } \kappa_2 = \kappa_1 + 2\nu\eta L_{2k}, \quad \sigma^2_{Fail|GD} = \max_{|\Omega| \leq 2k+K} \left[ 4\nu\eta\sqrt{k}\omega\mathbb{E}_{\mathcal{B}_t} \left[ \|\mathcal{P}_\Omega \left(\nabla f_{\mathcal{B}_t}\left(\mathbf{\Theta}^\star\right)\right)\|_F \right] \right].$$

**Remark 1.** *It is worth noting that $\sigma_{GD}, \sigma_{Fail|GD}$ are both statistical errors, which in the noiseless case are 0. In the case that the magnitude of top-$2k$ elements in the gradient are all greater than $\Delta$, we have $k_\Delta = 0$, which implies $\sigma_{\Delta|GD} = 0$. In this case ATEE's approximation doesn't incur any additional error compared with exact IHT.*

Theorem 3 shows that by setting $k = \Theta(KL_{2k}^2/\alpha_{2k}^2), \eta = \alpha_{2k}/2L_{2k}^2$, the parameter estimation can be improved geometrically when ATEE succeeds. We will show in Theorem 5 that with suffciently large batch size $m$, $\alpha_{2k}, L_{2k}$ are controlled and don't scale with $k, p$. When ATEE fails, it can't make the $\mathbf{\Theta}$ estimation worse by too much. Given that success rate of ATEE is controlled in Theorem 1, it naturally suggests that we can obtain the linear convergence in expectation. This leads to Theorem 4.

Define $\sigma_1^2 = \sigma_{GD}^2 + \sigma_{\Delta|GD}^2$, and $\sigma_2^2 = \sigma_{GD}^2 + \sigma_{Fail|GD}$. Let $\phi_t$ to be the success indicator of ATEE at time step $t$, and $\Phi_t = \{\phi_0, \phi_1, ..., \phi_t\}$. By Theorem 1, with $d = 48\log 2ck$, ATEE recovers top-$2k$ with probability at least $(1 - 1/c)$, we can easily show the convergence of Algorithm 1 as

**Theorem 4** (Main result). *Following the above notations, the expectation of the parameter recovery error of Algorithm 1 is bounded by*

$$\mathbb{E}_{B_t, \Phi_t} \left[ \|\mathbf{\Theta}^t - \mathbf{\Theta}^\star\|_F^2 \right] \leq \left( \kappa_1 + \frac{1}{c}(\kappa_2 - \kappa_1) \right)^t \|\mathbf{\Theta}^0 - \mathbf{\Theta}^\star\|_F^2$$

$$+ \left[ \left( \kappa_1 + \frac{1}{c}(\kappa_2 - \kappa_1) \right)^t - 1 \right] \left( \frac{\sigma_1^2}{\kappa_1 - 1} \right) + \frac{\kappa_2 - 1}{c - c\kappa_1 + \kappa_1 - \kappa_2} \left( \frac{\sigma_2^2}{\kappa_2 - 1} - \frac{\sigma_1^2}{\kappa_1 - 1} \right).$$

This shows that Algorithm 1 achieves linear convergence by setting $c \geq (\kappa_2 - \kappa_1)/(1 - \kappa_1)$. With $c$ increasing, the error ball converges to $\sigma_1^2/(1 - \kappa_1)$. The proof follows directly by taking expectation of the result we obtain in Theorem 3 with the recovery success probability established in Theorem 1.

**Computational analysis** With the linear convergence, the computational complexity is dominated by the complexity per iteration. Before discussing the complexity, we first establish the dependency between $L_k, \alpha_k$ and $m$ in the special case of quadratic regression, where the link function is identity. Notice that similar results would hold for more general quadratic problems as well.

**Theorem 5** (Minimum batch size). *For feature vector $\mathbf{x} \in \mathbb{R}^p$, whose first $p - 1$ coordinates are drawn i.i.d. from a bounded distribution, and the $p$-th coordinate is constant 1. W.l.o.g., we assume the first $p - 1$ coordinates to be zero mean, variance 1 and bounded by $B$. With batch size $m \gtrsim kB\log p/\epsilon^2$ we have $\alpha_k \geq 1 - \epsilon$, $L_k \leq 1 + \epsilon$ with high probability.*

Note that the sample complexity requirement matches the known information theoretic lower bound for recovering $k$-sparse $\mathbf{\Theta}$ up to a constant factor. The proof is similar to the analysis of restricted isometry property in sparse recovery. Recall that by Theorem 1, we have the per-iteration complexity $\widetilde{\mathcal{O}}(m(p+b))$. Combining the results of Lemma 2, Theorems 4 and 5, we have the following corollary on the complexity:

**Corollary 6** (Achieving sub-quadratic space and time complexity). *In the case of quadratic regression, by setting the parameters as above, IntHT recovers $\mathbf{\Theta}^\star$ in expectation up to a noise ball with linear convergence. The time and space complexity of IntHT is $\widetilde{\mathcal{O}}(k(k+p))$, which is sub-quadratic when $k$ is $\mathcal{O}(p^\gamma)$ for $\gamma < 1$.*

Note that the optimal time and space complexity is $\Omega(kp)$, since a minimum of $\Omega(k)$ samples are required for recovery, and $\Omega(p)$ for reading all entries. Corollary 6 shows the time and space complexity of IntHT is $\widetilde{\mathcal{O}}(k(k+p))$, which is nearly optimal.

## 5 Synthetic Experiments

To examine the sub-quadratic time and space complexity, we design three tasks to answer the following three questions: (i) Whether Algorithm 1 maintains linear convergence despite the hard thresholding not being accurate? (ii) What is the dependency between $b$ and $k$ to guarantee successful recovery? (iii) What is the dependency between $m$ and $p$ to guarantee successful recovery? Recall that

the per-iteration complexity of Algorithm 1 is $\widetilde{O}(m(p+b))$, where $b$ upper bounds the size of ATEE's output set, $p$ is the dimension of features and $m$ is batch size and $k$ is the sparsity of estimation. It will be clear as we proceed how the three questions can support sub-quadratic complexity.

**Experimental setting** We generate feature vectors $\mathbf{x}_i$, whose coordinates follow i.i.d. uniform distribution on $[-1, 1]$. Constant 1 is appended to each feature vector to model the linear terms and intercept. The true support is uniformly selected from all the interaction and linear terms, where the non-zero parameters are then generated uniformly on $[-20, -10] \cup [10, 20]$. Note that for the experiment concerning minimum batch size $m$, we instead use Bernoulli distribution to generate both the features and the parameters, which reduces the variance for multiple random runs and makes our phase transition plot clearer. The output $y_i$s, are generated following $\mathbf{x}_i^\top \mathbf{\Theta}^\star \mathbf{x}_i$. On the algorithm side, by default, we set $p = 200$, $d = 3$, $K = 20$, $k = 3K$, $\eta = 0.2$. Support recovery results with different $b$-$K$ combinations are averaged over 3 independent runs, results for $m$-$p$ combinations are averaged over 5 independent runs. All experiments are terminated after 150 iterations.

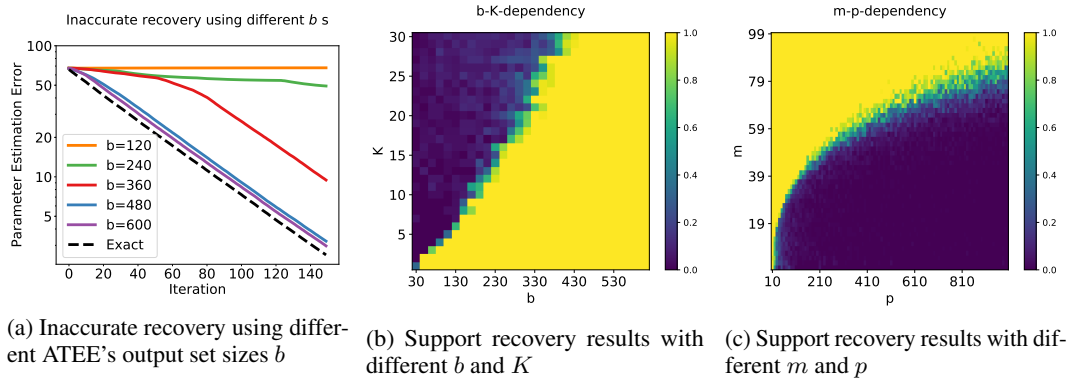

(a) Inaccurate recovery using different ATEE's output set sizes $b$

(b) Support recovery results with different $b$ and $K$

(c) Support recovery results with different $m$ and $p$

Figure 1: Synthetic experiment results: note $b, m$ are the parameters we used for IntHT and ATEE, where $b$ upper bounds the size of ATEE's output set and $m$ is the batch size used for IntHT. Recall $p$ is the dimension of features and $K$ is the sparsity of $\mathbf{\Theta}^\star$. **(a)** the convergence behavior with different choices of $b$. Linear convergence holds for small $b$, e.g., 360, when the parameter space is around $20,000$. **(b)** Support recovery results with different choices of $(b, K)$. We observe a linear dependence between $b$ and $K$. **(c)** Support recovery results with different choices of $(m, p)$. $m$ scales sub-linearly with $p$ to ensure a success recovery.

**Inaccurate support recovery with different $b$'s** Figure 1-(a) demonstrates different convergence results, measured by $\|\mathbf{\Theta} - \mathbf{\Theta}^\star\|_F$ with multiple choices of $b$ for ATEE in Algorithm 1. The dashed curve is obtained by replacing ATEE with exact top elements extraction (calculates the gradient exactly and picks the top elements). This is statistically optimal, but comes with quadratic complexity. By choosing a moderately large $b$, the inaccuracy induced by ATEE has negligible impact on the convergence. Therefore, Algorithm 1 can maintain the linear convergence despite the support recovery in each iteration is inaccurate. This aligns with Theorem 3. With linear convergence, the per iteration complexity will dominate the overall complexity.

**Dependency between $b$ and sparsity $k$** We proceed to see the proper choice of $b$ under different sparsity $k$ (we use $k = 3K$). We vary the sparsity $K$ from 1 to 30, and apply Algorithm 1 with $b$ ranges from 30 to 600. As shown in Figure 1-(b), the minimum proper choice of $b$ scales no more than linearly with $k$. This agrees with our analysis in Theorem 1. The per-iteration complexity then collapse to $\widetilde{\mathcal{O}}(m(p+k))$.

**Dependency between batch size $m$ and dimension $p$** Finally, we characterize the dependency between minimum batch size $m$ and the input dimension $p$. This will complete our discussion on the per-iteration complexity. The batch size varies from 1 to 99, and the input dimension varies from 10 to 1000. In this experiment, we employ the Algorithm 1 with ATEE replaced by exact top-$k$ elements extraction. Figure 1-(c) demonstrates the support recovery success rate of each $(k, p)$ combination. It shows the minimum batch size scales in logarithm with dimension $p$, as we proved in Theorem 5. Together with the previous experiment, it establishes the sub-quadratic complexity.

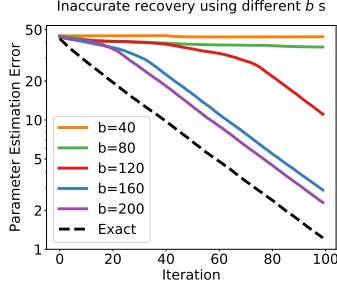

Figure 2: 3-order regression support recovery using different ATEE's output set sizes $b$

**Higher order interaction** IntHT is also extensible to higher order interactions. Specifically, by exploiting similar gradient structure $\sum r_i \mathbf{x}_i \otimes \mathbf{x}_i \otimes \mathbf{x}_i$, where $r_i$ denotes the residual for $(\mathbf{X}_i, y_i)$, $\otimes$ denotes the outer product of vector, we can again combine sketching with high-dimensional optimization to achieve nearly linear time and space (for constant sparsity).

For the experiment, we adopt the similar setting as for the **Inaccurate support recovery with different $b$s** experiment. The main difference is that we change from $y_i = \mathbf{x}_i^\top \Theta^\star \mathbf{x}_i$ to $y_i = \sum \Theta_{i,j,k} \mathbf{x}_i \mathbf{x}_j \mathbf{x}_k$, where $\Theta$ is now a three dimension tensor. Further, we set the dimension of $\mathbf{x}$ to $30$ and the sparsity $K = 20$. Figure 2 demonstrates the result of support recovering of 3-order interaction terms with different setting of $b$, where $b$ still bounds the size of ATEE's output set. We can see that IntHT still maintains the linear convergence in the higher order setting.

## Acknowledgement

We would like to acknowledge NSF grants 1302435 and 1564000 for supporting this research.

## Footnotes

[2]For now, we assume $L_{2k}$ to be a constant independent of $p, k$. We will discuss this in Theorem 5.

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
