[Supplementary Material]

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

(\boldsymbol{\Theta}_1) - F_n(\boldsymbol{\Theta}_2) \geq \langle \boldsymbol{\Theta}_1 - \boldsymbol{\Theta}_2, \nabla_{\boldsymbol{\Theta}} F_n(\boldsymbol{\Theta}_2) \rangle + \frac{\alpha_s}{2} \|\boldsymbol{\Theta}_1 - \boldsymbol{\Theta}_2\|_F^2$$

- *The mini-batch loss $f_{\mathcal{B}}$ satisfies $L_s$-Restricted Strong Smoothness (RSM):*

$$\|\nabla f_{\mathcal{B}}(\boldsymbol{\Theta}_1) - \nabla f_{\mathcal{B}}(\boldsymbol{\Theta}_2)\|_F \leq L_s \|\boldsymbol{\Theta}_1 - \boldsymbol{\Theta}_2\|_F, \, \forall \mathcal{B} \subset [n], \, |\mathcal{B}| = m$$

- *$f_{\mathcal{B}}$ satisfies Restricted Convexity (RC) (but not strong):*

$$f_{\mathcal{B}}(\boldsymbol{\Theta}_1) - f_{\mathcal{B}}(\boldsymbol{\Theta}_2) - \langle \nabla f_{\mathcal{B}}(\boldsymbol{\Theta}_2), \boldsymbol{\Theta}_1 - \boldsymbol{\Theta}_2 \rangle \geq 0, \, \forall \mathcal{B} \subset [n], \, |\mathcal{B}| = m, \, s = 3k + K$$

**Note:** While our assumptions are standard, our result does not follow immediately from existing analyses – because we cannot find the exact top elements of the gradient. We need to do a new analysis to show that even with our approximate top element extraction, linear convergence to $\boldsymbol{\Theta}^\star$ still holds.

## 4.2 Main Results

Here we proceed to establish the sub-quadratic complexity and consistency of IntHT for parameter estimation. Theorem 1 presents the analysis of ATEE. It provides the computation complexity analysis, as well as the statistical guarantee of support recovery. Based on this, we show the per round convergence property of Algorithm 1 in Theorem 3. We then establish our main statistical result, the linear convergence of Algorithm 1 in Theorem 4.

Next, we discuss the batch size that guarantees support recovery in Theorem 5, focusing on the quadratic regression setting, i.e. the model is linear in both interaction terms and linear terms. Combining all the established results, the sub-quadratic complexity is established in Corollary 6. All the proofs in this subsection can be found in Appendix E.

**Analysis of ATEE** Consider ATEE with parameters set to be $b, d, \Delta$. Recall this means that ATEE returns an index set $(\widetilde{\Lambda})$ of size at most $b$, which is expected to contain the desired index set $(\Lambda)$. Note that the desired index set $(\Lambda)$ is composed by the top-$2k$ elements of gradient $\nabla f_{\mathcal{B}}(\boldsymbol{\Theta})$ whose absolute value is greater than $\Delta$. Suppose now the current estimate is $\boldsymbol{\Theta}$, and $\mathcal{B}$ is the batch. The following theorem establishes when this output set $(\widetilde{\Lambda})$ captures the top elements of the gradient.

**Theorem 1** (Recovering top-$2k$ elements of the gradient, modified from [21])**.** *With the setting above, if we choose $b, d, \Delta$ so that $b\Delta^2 \geq 432 \left\|\nabla f_{\mathcal{B}}(\boldsymbol{\Theta})\right\|_F^2$ and $d \geq 48 \log 2ck$, then the index set $(\widetilde{\Lambda})$ returned by ATEE contains the desired index set $(\Lambda)$ with probability at least $1 - 1/c$.*

*Also in this case the time complexity of ATEE is $\widetilde{\mathcal{O}}\left(m(p+b)\right)$, and space complexity is $\widetilde{\mathcal{O}}\left(m(p+b)\right)$.*

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

# A   Details of ATEE

In this section, we provide the formal algorithm for ATEE, stated in Algorithm 3. ATEE consists of two sub-routines: an efficient sketching operation (line 5-11), and an efficient extraction operation (line 12-15).

For the sketching part, the algorithm first generate expander code which maps $[p]$ to $\{0,1\}^l$, where $l$ is the length of the codeword. Based on this encoding, we construct a table $\mathbf{E} \in \{0,1\}^{p \times l}$ where the $i$-th row is the codeword which encoded from $i$. Denote $\mathbf{e}_r$ as the $r$-th column of $\mathbf{E}$, we construct diagonal matrix $\mathbf{I}_{\mathbf{e}_r} = \text{diag}(\mathbf{e}_r)$. Then $2l$ different sub-matrices of $\mathbf{A}$ and $\mathbf{B}$ is constructed by $\mathbf{C}_r = \mathbf{I}_{\mathbf{e}_r}\mathbf{A}, \mathbf{C}_{r+l} = \mathbf{I}_{\mathbf{e}_r}\mathbf{B}, \forall r \in [l]$. It then sketches $\mathbf{A}\mathbf{C}_{r+l}^\top$ and $\mathbf{C}_r\mathbf{B}$, each of the matrices into a $b$ length vector, where $b \ll p^2$. The result is stored in $\mathbf{S} \in \mathbb{R}^{2l \times b}$. By exploiting the factorization of this matrix, the matrix outer product can be sketched in $\mathcal{O}(b \log(b))$ using fast Fourier transform (FFT) as used in [21](line 18 - 21).

For the sub-linear extraction, we first binarify the matrix $\mathbf{S}$ with threshold $\Delta/2$. Then each column of $\mathbf{S}$ becomes a codeword, where the first $l$ bits encodes the row index of the elements whose absolute value is greater than $\Delta/2$, the last $l$ bits encode its column index. By using expander code, it takes linear time $\mathcal{O}(l)$ to finish decoding. The whole process will be repeated for $d$ times and only the elements that are recovered for more than $d/2$ times will be recorded for output. This can boost the success probability of top-$k$ support recovery.

---

**Algorithm 3** APPROXIMATE TOP ELEMENTS EXTRACTION (ATEE-FORMAL)

---

1: **Input:** Matrix $\mathbf{A}$, matrix $\mathbf{B}$, top selection size $k$
2: **Parameters:** Output set size limit $b$, repetition number $d$, significant level $\Delta$
3: **Expected Output:** A set $\Lambda$, which is the top-$k$ elements in $\mathbf{A}\mathbf{B}^\top$ whose absolute value is also greater than $\Delta$
4: **Output:** Set $\widetilde{\Lambda}$ of indices, with size at most $b$ and approximately contains $\Lambda$
5: **for** $t = 0$ **to** $d - 1$ **do**
6:     Construct expander code table $\mathbf{E} \in \{0,1\}^{p \times l}$. Let $\mathbf{e}_r$ be the $r$-th column of $\mathbf{E}$
7:     $\mathbf{I}_{\mathbf{e}_r} = \text{diag}(\mathbf{e}_r), \mathbf{C}_r = \mathbf{I}_{\mathbf{e}_r}\mathbf{A}, \mathbf{C}_{r+l} = \mathbf{I}_{\mathbf{e}_r}\mathbf{B}, \forall r \in [l]$. Init $\mathbf{S}$ as a $2l \times b$ matrix
8:     Generate pairwise independent hash functions $h_1, h_2 : [p] \to [b]$
9:     **for** $r = 0$ **to** $l - 1$ **do**
10:         $\mathbf{S}[r,:] = \text{COMPRESSED-PRODUCT}(\mathbf{C}_r, \mathbf{B}, b, h_1, h_2)$
11:         $\mathbf{S}[r+l,:] = \text{COMPRESSED-PRODUCT}(\mathbf{A}, \mathbf{C}_{r+l}, b, h_1, h_2)$
12:     **for** $q = 0$ **to** $b - 1$ **do**
13:         $\mathbf{o}^q = \mathbf{1}(\mathbf{S}[:,q] > \frac{\Delta}{2})$
14:         $(i, j) = \text{DECODE}(\mathbf{o}^q; \mathbf{E})$
15:         $S = S \cup \{(i, j)\}$
16: **Return:** $\{(i,j) | \#(i,j) \in S \geq \frac{d}{2}\}$

---

17: **function** COMPRESSED-PRODUCT$(\mathbf{A}, \mathbf{B}, b, h_1, h_2)$:
18:     Generate random sign functions $s_1, s_2 : [p] \to \{-1, +1\}$, column length of $\mathbf{A}, \mathbf{B}$ are $m$
19:     **for** $i = 0$ **to** $m - 1$ **do**
20:         $\mathbf{p}_{\mathbf{a}_i} \leftarrow \text{COUNT-SKETCH}(\mathbf{a}_i, h_1, s_1, b), \mathbf{p}_{\mathbf{b}_i} \leftarrow \text{COUNT-SKETCH}(\mathbf{b}_i, h_2, s_2, b)$
21:         $\mathbf{s}_i = \text{IFFT}\left(\text{FFT}\left(\mathbf{p}_{\mathbf{a}_i}\right) \circ \text{FFT}\left(\mathbf{p}_{\mathbf{b}_i}\right)\right)$
22:     **Return:** $\mathbf{s} = \sum_{i=0}^{m-1} \mathbf{s}_i$

---

23: **function** COUNT-SKETCH$(\mathbf{x}, h, s, b)$
24:     Init $\mathbf{p}$ as a length $b$ vector
25:     **for** $i = 0$ **to** $p - 1$ **do**
26:         $\mathbf{p}[h(i)] = \mathbf{p}[h(i)] + s(i)\mathbf{x}[i]$
27:     **Return:** $\mathbf{p}$

---

## B IntHT-VR Algorithm

The application of IntHT to SVRG follows a very similar path as we apply it to SGD. The only trick is to ultilize the linearity of sketching. For SVRG, we will generate the hash function $h_1, h_2, s_1, s_2$ as described in SGD case at the begining of each outer iteration. They will be kept same through all the inner iterations. The skeching result $\mathbf{s}^i$ of the full gradient at the beginning of $i$-th outer loop will add up with the sketching result $\widetilde{\mathbf{s}}_j^i$ of the corresponding $j$-th inner loop. The summation $\mathbf{s}_j^i$ then goes through the decoding process, which is the same as SGD.

---

**Algorithm 4** INTHT WITH VARIANCE REDUCTION (INTHT-VR)

1: **Input:** Dataset $\{\mathbf{x}_{i'}, y_{i'}\}_{i'=0}^{n-1}$, threshold $\Delta$, dimension $p$, outer / inner round number $T/t$
2: **Output:** $\widehat{\Theta}$
3: **Parameters:** Codeword length $l$, sketch size $b$, repetition number $d$
4: Initialize $\Theta^0$ as a $p \times p$ zero matrix.
5: **for** $i = 0$ **to** $T - 1$ **do**
6: $\quad$ $\Theta_0^i = \Theta^i$, construct expander code table $\mathbf{E}^i \in \{0,1\}^{p \times l}$.
7: $\quad$ Generate hash functions $h_1, h_2 : [p] \rightarrow [b]$, and $s_1, s_2 : [p] \rightarrow \{-1, 1\}$
8: $\quad$ $\mathbf{s}^i = \text{INTERACTION-SKETCH}(\{\mathbf{x}_{i'}, u_{i'}\}_{i'=0}^{n-1}, \mathbf{E}^i, h_1, h_2, s_1, s_2)$
9: $\quad$ $\mathbf{G}^i := \frac{1}{n} \sum_{i'=0}^{n-1} u_{i'} \mathbf{x}_{i'} \mathbf{x}_{i'}^\top$
10: $\quad$ **for** $j = 0$ **to** $t - 1$ **do**
11: $\quad\quad$ Randomly pick a sample (min-batch) $\mathcal{B}_j^i$
12: $\quad\quad$ $\widetilde{u}_{i'} = u_{i'}(\Theta_j^i, \mathbf{x}_{i'}, y_{i'}) - u_{i'}(\Theta_0^i, \mathbf{x}_{i'}, y_{i'}), \forall i' \in \mathcal{B}_j^i, \widetilde{\mathbf{G}}_j^i := \frac{1}{|\mathcal{B}_j^i|} \sum_{i \in \mathcal{B}_j^i} \widetilde{u}_{i'} \mathbf{x}_{i'} \mathbf{x}_{i'}^\top$
13: $\quad\quad$ $\widetilde{\mathbf{s}}_j^i = \text{INTERACTION-SKETCH}(\{\mathbf{x}_{i'}, \widetilde{u}_{i'}\}_{i' \in \mathcal{B}_j^i}, \mathbf{E}^i, h_1, h_2, s_1, s_2)$
14: $\quad\quad$ $\mathbf{s}_j^i = \mathbf{s}^i + \widetilde{\mathbf{s}}_j^i$
15: $\quad\quad$ $\widetilde{S}_j^i = \text{INTERACTION-DECODE}(\mathbf{s}_j^i, \Delta, \mathbf{E}^i)$
16: $\quad\quad$ $S_j^i = \widetilde{S}_j^i \cup \text{supp}(\Theta_j^i)$
17: $\quad\quad$ $\mathcal{P}_{S_j^i}(\mathbf{G}_j^i) \leftarrow$ the gradient value $\mathbf{G}_j^i := \mathbf{G}^i + \widetilde{\mathbf{G}}_j^i$ calculated only on index set $S_j^i$,
18: $\quad\quad$ $\Theta_{j+1}^i = \mathcal{H}_k \left( \Theta_j^i - \eta \mathcal{P}_{S_j^i}(\mathbf{G}_j^i) \right)$
19: $\quad$ $\Theta^{i+1} = \Theta_{j'}^i$, for $j' \sim \text{Unif}(\{0, \cdots, t-1\})$
20: **Return:** $\widehat{\Theta} = \Theta^T$

---

21: **function** INTERACTION-SKETCH$(\{\mathbf{x}_i, u_i\}_{i \in \mathcal{B}}, \mathbf{E}, h_1, h_2, s_1, s_2)$ :
22: $\quad$ Set $\mathbf{A} \in \mathbb{R}^{p \times |\mathcal{B}|}$, where each column of $\mathbf{A}(\mathbf{B})$ is $u_i \mathbf{x}_i, i \in \mathcal{B}$.
23: $\quad$ Set $\mathbf{B} \in \mathbb{R}^{p \times |\mathcal{B}|}$, where each column of $\mathbf{A}(\mathbf{B})$ is $\mathbf{x}_i, i \in \mathcal{B}$.
24: $\quad$ Set $\mathbf{s}$ as a $d \times 2l \times b$ tensor.
25: $\quad$ **for** $t = 0$ **to** $d - 1$ **do**
26: $\quad\quad$ Let $\mathbf{e}_r$ be the $r$-th column of $\mathbf{E}$, $\mathbf{I}_{\mathbf{e}_r} = \text{diag}(\mathbf{e}_r)$, $\mathbf{C}_r = \mathbf{I}_{\mathbf{e}_r} \mathbf{A}$, $\mathbf{C}_{r+l} = \mathbf{I}_{\mathbf{e}_r} \mathbf{B}, \forall r \in [l]$.
27: $\quad\quad$ **for** $r = 0$ **to** $l - 1$ **do**
28: $\quad\quad\quad$ $\mathbf{s}[t, r, :] = \text{COMPRESSED-PRODUCT}(\mathbf{C}_r, \mathbf{B}, b, h_1, h_2)$
29: $\quad\quad\quad$ $\mathbf{s}[t, r + l, :] = \text{COMPRESSED-PRODUCT}(\mathbf{A}, \mathbf{C}_{r+l}, b, h_1, h_2)$
30: $\quad$ **Return: s**

---

31: **function** INTERACTION-DECODE$(\mathbf{s}, \Delta, \mathbf{E})$ :
32: $\quad$ Set $S = \emptyset$
33: $\quad$ **for** $t = 0$ **to** $d - 1$ **do**
34: $\quad\quad$ **for** $q = 0$ **to** $b - 1$ **do**
35: $\quad\quad\quad$ $\mathbf{o}^{t,q} = \mathbf{1}(\mathbf{s}^{t,:,q} > \frac{\Delta}{2})$
36: $\quad\quad\quad$ $(i, j) = \text{DECODE}(\mathbf{o}^{t,q}, \mathbf{E})$
37: $\quad\quad\quad$ $S = S \cup \{(i, j)\}$
38: $\quad$ **Return:** $\{(i, j) | \#(i, j) \in S \geq \frac{d}{2}\}$

---

## C   IntHT-VR Analysis

Here we proceed to provide theoretical guarantee for Appendix B.

Similar to the definitions for Theorem 3 and Theorem 4, we define $\Lambda_{2k}$ to be the set of top $2k$ elements in $\mathbf{G}_{j+1}^i$, $\Lambda_\Delta$ to be the set of elements in $\mathbf{G}_{j+1}^i$ whose magnitude is greater than $\Delta$ and the output set of ATEE to be $\widetilde{\Lambda}$. We have the support of interest $\Lambda = \Lambda_{2k} \cap \Lambda_\Delta$ and the number of top-$2k$ elements whose magnitude is below $\Delta$ $k_\Delta = |\Lambda_{2k} \backslash \Lambda_\Delta|$. Define $\mathcal{B}_i$ to be the set of samples used during $i$-th outer loop. Recall that $\eta$ is the step size and $m$ is the batch size. We then have the following result:

**Theorem 7** (Per-round Convergence of Algorithm 4). *If $\Lambda \subseteq \widetilde{\Lambda}$, the per-round convergence of Algorithm 4 is as follows:*

$$\mathbb{E}_{\mathcal{B}_i}\left[F(\mathbf{\Theta}_0^{i+1}) - F(\mathbf{\Theta}^\star))\right] \leq \kappa_{SVRG}\left[F(\mathbf{\Theta}_0^i) - F(\mathbf{\Theta}^\star)\right] + \sigma_{SVRG}$$

*where $\nu = 1 + \frac{\rho + \sqrt{(4+\rho)\rho}}{2}, \rho = K/k$, $t$ is the inner round number, and*

$$\kappa_{SVRG} = \frac{1}{\alpha_{2k}\nu\eta(1 - 2\eta L_{2k})t} + \frac{2\eta L_{2k}}{1 - 2\eta L_{2k}},$$

$$\sigma_{SVRG} = \frac{4\nu\eta\sigma'(4L_{2k}\sqrt{k}\omega + \sigma')t + \frac{2}{\alpha_{2k}}\sqrt{k}\omega\sigma' + t\sigma_{\Delta|SVRG}^2}{2\nu\eta(1 - 2\eta L_{2k})t},$$

$$\sigma' = \max_{|\Omega| = 3k+K}\|\mathcal{P}_\Omega F(\mathbf{\Theta}^*)\|_F, \quad \sigma_{\Delta|SVRG}^2 = 4\sqrt{k_\Delta}\eta\sqrt{k}\omega\Delta + 2k_\Delta\eta^2\Delta^2.$$

*To ensure the convergence, it requires that*

$$\eta < \frac{1}{4L_{2k}}, \quad \nu < \frac{1}{1 - \eta\alpha_{2k}}.$$

The proof can be found in Appendix F.1.

**Remark 2.** *Similar to the Remark 1 case, $\sigma'$ is statistical error, which in noiseless case are 0. In the case that the magnitude of top-$2k$ elements in the gradient are all greater than $\Delta$, we have $\Lambda_{2k} \subseteq \Lambda_\Delta$ and $k_\Delta = 0$, which implies $\sigma_{\Delta|SVRG} = 0$.*

To obtain the convergence result over all iterations, we adopt the same definition and assumption as in Theorem 7. By setting $c = \Theta(p)$, $d = 48\log(ck)$, we have that the inner loop of Algorithm 4 succeeds with high probability (recall that $c$ was used to control the failure probability in Theorem 1, and it is not hard to see that the property in Theorem 1 still holds for IntHT-VR). Then we have the following result:

**Theorem 8** (Convergence of IntHT-VR). *Under the same parameter setting as in Theorem 1, with $d$ specifically defined as above, the convergence of Algorithm 4 is given by*

$$\mathbb{E}_{B_t}\left[F(\mathbf{\Theta}_0^t) - F(\mathbf{\Theta}^\star))\right] \leq \kappa_{SVRG}^s\left[F(\mathbf{\Theta}_0^0) - F(\mathbf{\Theta}^\star)\right] + \frac{\sigma_{SVRG}}{1 - \kappa_{SVRG}},$$

*where the definitions of $\kappa_{SVRG}$ and $\sigma_{SVRG}$ follows from Theorem 7.*

*Proof.* Given that ATEE succeeds with high probability, the contraction of each iteration is characterized by Theorem 7. By solving the recursion, we have the desired convergence. ∎

**Remark 3.** *Here we set ATEE to succeed with high probability, where in Theorem 4 it only requires ATEE to succeed with constant probability. This is because in each inner loop, the iterations share the same hash function $s, h$ as specified in ATEE, which removes the independence of ATEE for each iteration. Intuitively, once ATEE fails, it could fail on the entire inner loop and ruined the estimation for $\mathbf{\Theta}$. By setting $c = \Theta(p)$, the high probability statement can be obtained without incurring more than $\mathcal{O}(\log p)$ factor higher complexity.*

# D Technical Lemmas and Corollaries

**Lemma 9** (Tight Bound for Hard Thresholding [26]). *Let $\mathbf{B} \in \mathbb{R}^{p \times p}$ be an arbitrary matrix and $\boldsymbol{\Theta} \in \mathbb{R}^{p \times p}$ be any $K$-sparse signal. For any $k \geq K$, we have the following bound:*

$$\|\mathcal{H}_k(\mathbf{B}) - \boldsymbol{\Theta}\|_F \leq \sqrt{\nu}\, \|\mathbf{B} - \boldsymbol{\Theta}\|_F, \quad \nu = 1 + \frac{\rho + \sqrt{(4+\rho)\rho}}{2}, \quad \rho = \frac{\min\{K, p^2 - k\}}{k - K + \min\{K, p^2 - k\}}$$

The provide a short proof in Appendix G.1.

**Corollary 10** (similar to co-coercivity). *For a given support set $\Omega$, assume that the continuous function $f(\cdot)$ is $L_{|\Omega|}$-RSS and $K$-RC. Then, for all matrices $\boldsymbol{\Theta}, \boldsymbol{\Theta}'$ with $|\mathrm{supp}(\boldsymbol{\Theta} - \boldsymbol{\Theta}') \cup \Omega| \leq K$,*

$$\|\mathcal{P}_\Omega\left(\nabla f(\boldsymbol{\Theta}') - \nabla f(\boldsymbol{\Theta})\right)\|_F^2 \leq 2L_{|\Omega|}\left(f(\boldsymbol{\Theta}') - f(\boldsymbol{\Theta}) - \langle \nabla f(\boldsymbol{\Theta}), \boldsymbol{\Theta}' - \boldsymbol{\Theta}\rangle\right).$$

The proof can be found in Appendix G.2.

**Corollary 11** (bounding $\|\mathcal{P}_\Omega(\mathbf{G}^t)\|_2^2$). *Let $\Omega = \mathrm{supp}(\boldsymbol{\Theta}^{t-1}) \cup \mathrm{supp}(\boldsymbol{\Theta}^t) \cup \mathrm{supp}(\boldsymbol{\Theta}^\star)$. For SGD and SVRG, we have the following bound:*

1. *SGD:* $\mathbf{G}^t = \nabla f_{\iota_t}\left(\boldsymbol{\Theta}^{t-1}\right)$

$$\mathbb{E}_{\iota_t}\left\|\mathcal{P}_\Omega(\mathbf{G}^t)\right\|_F^2 \leq 2L_{2k}^2\left\|\boldsymbol{\Theta}^{t-1} - \boldsymbol{\Theta}^\star\right\|_F^2 + 2\left\|\mathcal{P}_\Omega\left(\nabla f_{\iota_t}(\boldsymbol{\Theta}^\star)\right)\right\|_F^2$$

2. *SVRG:* $\mathbf{G}_j^i = \nabla f_{b_j}\left(\boldsymbol{\Theta}_j^i\right) - \nabla f_{b_j}\left(\boldsymbol{\Theta}_0^i\right) + \nabla F\left(\boldsymbol{\Theta}_0^i\right)$

$$\mathbb{E}_{b_j}\left\|\mathcal{P}_\Omega(\mathbf{G}_j^i)\right\|_F^2 \leq 4L_{2k}\left[F(\boldsymbol{\Theta}_j^i) - F(\boldsymbol{\Theta}^\star)\right] + 4L_{2k}\left[F(\boldsymbol{\Theta}_0^i) - F(\boldsymbol{\Theta}^\star)\right]$$
$$- 4L_{2k}\left\langle \nabla F(\boldsymbol{\Theta}^\star), \boldsymbol{\Theta}_j^i + \boldsymbol{\Theta}_0^i - 2\boldsymbol{\Theta}^\star\right\rangle + 4\left\|\mathcal{P}_\Omega(\nabla F(\boldsymbol{\Theta}^\star))\right\|_F$$

The proof can be found in Appendix G.3.

**Corollary 12** (HT property). *Let $\Lambda_{2k}$ be the support of the top-$2k$ entries in $\mathbf{G}$ with largest absolute value, for a $k$-sparse matrix $\boldsymbol{\Theta}$,*

$$\mathcal{H}_k\left(\boldsymbol{\Theta} - \eta\mathbf{G}\right) = \mathcal{H}_k\left(\boldsymbol{\Theta} - \eta\mathcal{P}_{\mathrm{supp}(\boldsymbol{\Theta}) \cup \Lambda_{2k}}(\mathbf{G})\right)$$

The proof can be found in Appendix G.4.

**Corollary 13** ($\Delta$-Inexact Hard Thresholding). *Define $\Lambda_\Delta$ to be the set of elements in $\mathbf{G}_t$ whose magnitude is greater than $\Delta$. Further define $\Lambda = \Lambda_{2k} \cap \Lambda_\Delta, k_\Delta = |\Lambda_{2k} \backslash \Lambda_\Delta|$. Define,*

$$\widetilde{\boldsymbol{\Theta}}^+ = \mathcal{H}_k\left(\boldsymbol{\Theta} - \eta\mathbf{G}_t\right)$$
$$\boldsymbol{\Theta}^+ = \mathcal{H}_k\left(\boldsymbol{\Theta} - \eta\mathcal{P}_{\widetilde{\Lambda} \cup \mathrm{supp}(\boldsymbol{\Theta})}(\mathbf{G}_t)\right)$$

*In the case $\Lambda_\Delta \subseteq \widetilde{\Lambda}$ and $\Lambda_\Delta \subseteq \widetilde{\Lambda}$, we have the bound,*

$$\|\boldsymbol{\Theta}^+ - \widetilde{\boldsymbol{\Theta}}^+\|_F \leq \eta\Delta\sqrt{2k_\Delta}$$

The proof can be found in Appendix G.5.

# E Proofs for Section 4

## E.1 Proof of Theorem 1

*Proof.* The proof of Theorem 1 heavily relies on the analysis in [21]. Given that $\nabla f_{\mathcal{B}_i}(\boldsymbol{\Theta})$ can be expressed as multiplication of two matrices, we slightly abuse the notation $\mathbf{A}, \mathbf{B}$ to denote the pair of matrices that $\mathbf{A}\mathbf{B}^\top = \nabla f_{\mathcal{B}_i}(\boldsymbol{\Theta})$.

Denote the output of COMPRESSED-PRODUCT as $\mathbf{s}$. Define the hash function $h_1, h_2 : [p] \to [b]$ and $s_1, s2 : [p] \to \{-1, 1\}$. Let $h$ be the hash function that satisfies $h(i, j) = h_1(i) + h_2(j) \mod b$, and $s$ be $s(i, j) = s_1(i)s_2(j)$. Let $\mathbf{1}_{i,j}^q$ be the indicator of event $\{h(i, j) = q\}$. Define the index set

of the top $2k$ elements (with largest abstract value) of $\mathbf{AB}^\top$ to be $\Psi_{2k}$. Denote the index set of the elements with absolute value greater than $\Delta$ as $\Psi_\Delta$. Let $\Psi = \Psi_{2k} \cap \Psi_\Delta$, and we are interested in finding all indices in $\Psi$.

Our proof consists of the following four main steps.

**Step I:** Bound the variance of a single decoded entry.

$$\mathbf{s}_q = \sum_{(i,j) \in [p] \times [p]} \mathbf{1}_{i,j}^q s(i,j) (\mathbf{AB}^\top)_{ij}$$

For $(i^\star, j^\star) \in \Psi$, with $q^\star = h(i^\star, j^\star)$ we have:

$$\mathbf{s}_{q^\star} = s(i^\star, j^\star)(\mathbf{AB}^\top)_{i^\star j^\star} + \sum_{(i,j) \neq (i^\star,j^\star),(i,j) \in [p] \times [p]} \mathbf{1}_{i,j}^{q^\star} s(i,j)(\mathbf{AB}^\top)_{ij}$$

Then,

$$
\begin{aligned}
|\mathbf{s}_{q^\star}| \geq & s(i^\star, j^\star) \operatorname{sign}\left((\mathbf{AB}^\top)_{i^\star j^\star}\right) \mathbf{s}_{q^\star} \\
= & \left|(\mathbf{AB}^\top)_{i^\star j^\star}\right| + s(i^\star, j^\star) \operatorname{sign}\left((\mathbf{AB}^\top)_{i^\star j^\star}\right) \sum_{(i,j) \neq (i^\star,j^\star),(i,j) \in [p] \times [p]} \mathbf{1}_{i,j}^{q^\star} s(i,j)(\mathbf{AB}^\top)_{ij}
\end{aligned}
$$

Let $\mathbf{s}'_{q^\star} = s(i^\star, j^\star) \operatorname{sign}\left((\mathbf{AB}^\top)_{i^\star j^\star}\right) \sum_{(i,j) \neq (i^\star,j^\star),(i,j) \in [p] \times [p]} \mathbf{1}_{i,j}^{q^\star} s(i,j)(\mathbf{AB}^\top)_{ij}$. We have:

$$\mathbb{P}\left(\text{ error in one bit }\right) \leq \mathbb{P}\left(|\mathbf{s}'_{q^\star}| \geq \frac{\Delta}{2}\right) \leq \frac{4\operatorname{var}(\mathbf{s}'_{q^\star})}{\Delta^2}$$

Taking expectation over all possible partitions (based on $h$), we have:

$$\mathbb{E}_h\left[\operatorname{var}(\mathbf{s}'_{q^\star})\right] = \frac{1}{b} \sum_{(i,j) \neq (i^\star,j^\star),(i,j) \in [p] \times [p]} (\mathbf{AB}^\top)_{ij}^2 \leq \frac{\left\|\mathbf{AB}^\top\right\|_F^2}{b}.$$

**Step II:** Bound the failure probability of recovering a single large entry.

By Markov's inequality, we have

$$\mathbb{P}\left(\operatorname{var}(\mathbf{s}'_{q^\star}) \geq \frac{c\left\|\mathbf{AB}^\top\right\|_F^2}{b}\right) \leq \frac{1}{c}.$$

Given the upper bound on $\operatorname{var}(\mathbf{s}'_{q^\star})$, which happens with probability at least $1 - \frac{1}{c}$ due to the randomness from $h$, the only left randomness comes from $s(i,j)$. Note that we use the same $h$ for every $t$. Then,

$$\mathbb{P}\left(|\mathbf{s}'_{q^\star}| \geq \frac{\Delta}{2}\right) \leq \frac{4c}{\Delta^2} \frac{\left\|\mathbf{AB}^\top\right\|_F^2}{b}$$

The above inequality gives an error bound for each bit in the error-correcting code. Thus for a length $l$ code, the expected number of wrong bits is:

$$\mathbb{E}\left[\text{ number of wrong bits }\right] = \sum_{r=0}^{2l-1} \mathbb{P}\left(\text{ the } r^{\text{th}} \text{ bit is wrong }\right) \leq \frac{4lc}{\Delta^2} \frac{\left\|\mathbf{AB}^\top\right\|_F^2}{b}$$

By using an expander code, we can tolerate a constant fraction of error $\delta$ which is independent of message length $\log p$, with a code length $l = \mathcal{O}(\log p)$ [28]. By Markov's inequality, and combining with the probability bound on $h$,

$$\mathbb{P}\left((i^\star, j^\star) \text{ not recovered }\right) \leq \mathbb{P}\left(\text{more than } \delta l \text{ wrong bits}\right) \leq \frac{4c}{\Delta^2} \frac{\left\|\mathbf{AB}^\top\right\|_F^2}{b\delta} + \frac{1}{c}$$

Optimizing over the constant $c$ (by setting $c = \frac{\Delta\sqrt{b\delta}}{2\|\mathbf{A}\mathbf{B}^\top\|_F}$), we have

$$\mathbb{P}\left((i^\star, j^\star) \text{ not recovered }\right) \leq \frac{4}{\Delta}\frac{\left\|\mathbf{A}\mathbf{B}^\top\right\|_F}{\sqrt{b\delta}}$$

By choosing $b \geq \frac{144\left\|\mathbf{A}\mathbf{B}^\top\right\|_F^2}{\Delta^2\delta}$, we have

$$\mathbb{P}\left((i^\star, j^\star) \text{ not recovered }\right) \leq \frac{1}{3}.$$

For simplicity, we take $\delta = \frac{1}{3}$. Combining the assumption that $\left\|\mathbf{A}\mathbf{B}^\top\right\|_F = \|\nabla f_{\mathcal{B}_i}(\boldsymbol{\Theta})\|_F$. Taking $\Delta \geq \|\nabla f_{\mathcal{B}}(\boldsymbol{\Theta})\|_F / \sqrt{2k}$, which implies $b \geq \frac{432\|\nabla f_{\mathcal{B}_i}(\boldsymbol{\Theta})\|_F^2}{\Delta^2}$ will give a constant probability to successfully recover $(i^\star, j^\star)$.

**Step III:** Union bound over all large entries. Repeat the count sketch and sub-linear extraction for $d$ times and take the $(i, j)$ pair that are recovered more than $d/2$ times, we have that

$$\mathbb{P}\left((i^\star, j^\star) \text{ not recovered for more than } d/2 \text{ times}\right) \leq \exp\left(-\frac{d}{48}\right).$$

Since the events of recovering different $(i, j) \in \Psi$ are not independent (because of the dependency induced by $h$ functions), we use union bound over all the elements in $\Psi$. Thus we have

$$\mathbb{P}\left(\Psi \text{ not recovered }\right) \leq |\Psi|\exp\left(-\frac{d}{48}\right) \leq 2k\exp\left(-\frac{d}{48}\right)$$

By taking $d = 48\log(2ck)$, we obtain the desired constant success rate $1 - \frac{1}{c}$ for recovering $\Psi$.

**Step IV:** For the overall time complexity of the Interaction Top Elements Extraction (ATEE), encoding the index will take $\mathcal{O}(pl)$. Each compressed product step will take $\mathcal{O}(mp + mb\log b)$ and it will be repeatedly calculated for $2l$ times, where $l$ is the length of the expander code. Given that expander code has a linear decoding complexity, thus the extraction step can be done with $\mathcal{O}(bl)$. The above mentioned procedure will be repeated for $d$ times. Putting everything together, we have the time complexity for ATEE is

$$\mathcal{O}\left(\underbrace{\log(ck)}_{\text{repeat d times}}\left[\underbrace{p\log(p) + b\log(p)}_{\text{encode \& decode}} + \underbrace{\log(p)\left[mp + mb\log(b)\right]}_{\text{compressed product}}\right]\right)$$

which achieves sub-quadratic time complexity. Ignoring the logarithm term, the time complexity is $\widetilde{\mathcal{O}}(m(p + b))$, which naturally implies that the space complexity is $\widetilde{\mathcal{O}}(m(p + b))$. ∎

### E.2   Proof of Lemma 2

*Proof.* By RSM, we have

$$\|\nabla f_{\mathcal{B}_t}(\boldsymbol{\Theta}) - \nabla f_{\mathcal{B}_t}(\boldsymbol{\Theta}^\star)\|_F \leq L_{2k}\|\boldsymbol{\Theta} - \boldsymbol{\Theta}^\star\|_F, \ \forall \boldsymbol{\Theta} \ s.t. \ |\mathrm{supp}(\boldsymbol{\Theta}) \cup \mathrm{supp}(\boldsymbol{\Theta}^\star)| \leq 2k$$

By triangle inequality,

$$\|\nabla f_{\mathcal{B}_t}\|_F \leq L_{2k}\|\boldsymbol{\Theta} - \boldsymbol{\Theta}^\star\|_F + \|\nabla f_{\mathcal{B}_t}(\boldsymbol{\Theta}^\star)\|_F$$

By the fact that $\|\boldsymbol{\Theta}\|_F \leq \sqrt{k}\omega$, the first term can be directly bounded by $L_{2k}\|\boldsymbol{\Theta} - \boldsymbol{\Theta}^\star\|_F \leq 2L_{2k}\sqrt{k}\omega$. For the last term we have $\|\nabla f_{\mathcal{B}_t}(\boldsymbol{\Theta}^\star)\|_F \leq G$. Thus we have,

$$\|\nabla f_{\mathcal{B}_t}(\boldsymbol{\Theta})\|_F \leq 2L_{2k}\sqrt{k}\omega + G$$

∎

### E.3 Proof of Theorem 3

*Proof.*
With stochastic gradient descent, we have $\mathbf{G}^t = \nabla f_{\mathcal{B}t}(\mathbf{\Theta}^{t-1})$ as the gradient at step $t$. The per-round convergence can be separately analyzed for the two cases.

**ATEE succeeds:** $\Lambda \subseteq \widetilde{\Lambda}$. Before analyzing $\mathbf{\Theta}^t$, we first construct an intermediate parameter $\widetilde{\mathbf{\Theta}}^t$ as,

$$\widetilde{\mathbf{\Theta}}^t = \mathcal{H}_k\left(\mathbf{\Theta}^{t-1} - \eta \mathcal{P}_{\Lambda_{2k} \cup \mathrm{supp}(\mathbf{\Theta}^{t-1})}\left(\mathbf{G}_t\right)\right) = \mathcal{H}_k\left(\mathbf{\Theta}^{t-1} - \eta \mathbf{G}_t\right)$$

The second inequality directly comes from Corollary 12. This is actually the best situation we can hope for. In this situation, the approximation projection in ATEE doesn't affect the update. We will start with the bound on $\left\|\widetilde{\mathbf{\Theta}}^t - \mathbf{\Theta}^\star\right\|_F^2$. $\mathbf{\Theta}^t$ will then be compared with $\widetilde{\mathbf{\Theta}}^t$ to obtain the error bound. We will never refer to $\widetilde{\mathbf{\Theta}}^t$ in practice, but this construction makes the proof much clear. Consider the proxy

$$\mathbf{Z}^t = \mathbf{\Theta}^{t-1} - \eta \mathbf{G}^t$$

Let $\Omega = \mathrm{supp}(\mathbf{\Theta}^{t-1}) \cup \mathrm{supp}(\widetilde{\mathbf{\Theta}}^t) \cup \mathrm{supp}(\mathbf{\Theta}^\star)$,

$$\left\|\widetilde{\mathbf{\Theta}}^t - \mathbf{\Theta}^\star\right\|_F^2 = \left\|\mathcal{H}_k\left(\mathbf{Z}^t\right) - \mathbf{\Theta}^\star\right\|_F^2 = \left\|\mathcal{H}_k\left(\mathcal{P}_\Omega\left(\mathbf{Z}^t\right)\right) - \mathbf{\Theta}^\star\right\|_F^2 \leq \nu \left\|\mathcal{P}_\Omega\left(\mathbf{Z}^t\right) - \mathbf{\Theta}^\star\right\|_F^2 \quad (3)$$

Notice that

$$\left\|\mathcal{P}_\Omega\left(\mathbf{Z}^t\right) - \mathbf{\Theta}^\star\right\|_F^2 \quad (4)$$

$$= \left\|\mathbf{\Theta}^{t-1} - \mathbf{\Theta}^\star - \eta \mathcal{P}_\Omega(\mathbf{G}^t)\right\|_F^2$$

$$\leq \left\|\mathbf{\Theta}^{t-1} - \mathbf{\Theta}^\star\right\|_F^2 + \eta^2 \left\|\mathcal{P}_\Omega(\mathbf{G}^t)\right\|_F^2 - 2\eta \left\langle \mathbf{\Theta}^{t-1} - \mathbf{\Theta}^\star, \mathbf{G}^t\right\rangle \quad (5)$$

Notice that $\mathbb{E}[\mathbf{G}^t] = \nabla F(\mathbf{\Theta}^{t-1})$. Equation (5) includes three terms: (i) the first term is the contraction term which will be kept, (ii) the second term is controlled by first using Corollary 11 then taking expectation, and (iii) the third term is controlled by first taking the expectation and then using the RSC property. Therefore,

$$\mathbb{E}_{\mathcal{B}_t}\left[\left\|\widetilde{\mathbf{\Theta}}^t - \mathbf{\Theta}^\star\right\|_F^2\right]$$

$$\leq \mathbb{E}_{\mathcal{B}_t}\left[\nu \left\|\mathbf{\Theta}^{t-1} - \mathbf{\Theta}^\star\right\|_F^2 + \nu\eta^2 \left\|\mathcal{P}_\Omega(\mathbf{G}^t)\right\|_F^2 - 2\nu\eta \left\langle \mathbf{\Theta}^{t-1} - \mathbf{\Theta}^\star, \mathbf{G}^t\right\rangle\right]$$

$$\overset{(a)}{\leq} \nu \left\|\mathbf{\Theta}^{t-1} - \mathbf{\Theta}^\star\right\|_F^2 - 2\nu\eta \left\langle \mathbf{\Theta}^{t-1} - \mathbf{\Theta}^\star, \nabla F\left(\mathbf{\Theta}^{t-1}\right)\right\rangle$$
$$+ \nu\eta^2 \left[2L_{2k}^2 \left\|\mathbf{\Theta}^{t-1} - \mathbf{\Theta}^*\right\|_F^2 + 2\mathbb{E}_{\mathcal{B}_t}\left[\left\|\mathcal{P}_\Omega \nabla f_{\mathcal{B}_t}\left(\mathbf{\Theta}^*\right)\right\|_F^2\right]\right]$$

$$= \nu \left\|\mathbf{\Theta}^{t-1} - \mathbf{\Theta}^\star\right\|_F^2 + \nu\eta^2 \left[2L_{2k}^2 \left\|\mathbf{\Theta}^{t-1} - \mathbf{\Theta}^*\right\|_F^2 + 2\mathbb{E}_{\mathcal{B}_t}\left[\left\|\mathcal{P}_\Omega \nabla f_{\mathcal{B}_t}\left(\mathbf{\Theta}^*\right)\right\|_F^2\right]\right]$$
$$- 2\nu\eta \left\langle \mathbf{\Theta}^{t-1} - \mathbf{\Theta}^\star, \nabla F\left(\mathbf{\Theta}^{t-1}\right) - \nabla F\left(\mathbf{\Theta}^\star\right)\right\rangle + 2\nu\eta \left\langle \mathbf{\Theta}^{t-1} - \mathbf{\Theta}^\star, \nabla F\left(\mathbf{\Theta}^\star\right)\right\rangle$$

$$\overset{(b)}{\leq} \nu \left\|\mathbf{\Theta}^{t-1} - \mathbf{\Theta}^\star\right\|_F^2 + \nu\eta^2 \left[2L_{2k}^2 \left\|\mathbf{\Theta}^{t-1} - \mathbf{\Theta}^*\right\|_F^2 + 2\mathbb{E}_{\mathcal{B}_t}\left[\left\|\mathcal{P}_\Omega \nabla f_{\mathcal{B}_t}\left(\mathbf{\Theta}^*\right)\right\|_F^2\right]\right]$$
$$- 2\nu\eta\alpha_{2k} \left\|\mathbf{\Theta}^{t-1} - \mathbf{\Theta}^\star\right\|_F^2 + 2\nu\eta \left\|\mathbf{\Theta}^{t-1} - \mathbf{\Theta}^\star\right\|_F \left\|\mathcal{P}_\Omega \nabla F\left(\mathbf{\Theta}^\star\right)\right\|_F$$

$$= \nu \left(1 - 2\eta\alpha_{2k} + 2\eta^2 L_{2k}^2\right) \left\|\mathbf{\Theta}^{t-1} - \mathbf{\Theta}^\star\right\|_F^2$$
$$+ 2\nu\eta \left\|\mathbf{\Theta}^{t-1} - \mathbf{\Theta}^\star\right\|_F \left\|\mathcal{P}_\Omega\left(\nabla F\left(\mathbf{\Theta}^\star\right)\right)\right\|_F + 2\nu\eta^2 \mathbb{E}_{\mathcal{B}_t} \left\|\mathcal{P}_\Omega\left(\nabla f_{\mathcal{B}_t}\left(\mathbf{\Theta}^\star\right)\right)\right\|_F^2$$

where the first inequality is due to Equation (3) and Equation (5). (a) plugs in the result from Corollary 11 and takes expectation over the gradient. (b) uses RSC property and Cauchy-Shwartz inequality.

Suppose each coordinate of $\mathbf{\Theta}$ is bounded by $\omega$, we know that $\left\|\mathbf{\Theta}^{t-1}\right\|_F \leq \sqrt{k}\omega$ and $\left\|\mathbf{\Theta}^\star\right\|_F \leq \sqrt{k}\omega$, we further have

$$\mathbb{E}_{\mathcal{B}_t}\left[\left\|\widetilde{\mathbf{\Theta}}^t - \mathbf{\Theta}^\star\right\|_F^2\right] \leq \nu \left(1 - 2\eta\alpha_{2k} + 2\eta^2 L_{2k}^2\right) \left\|\mathbf{\Theta}^{t-1} - \mathbf{\Theta}^\star\right\|_F^2$$
$$+ 4\nu\eta\sqrt{k}\omega \left\|\mathcal{P}_\Omega\left(\nabla F\left(\mathbf{\Theta}^\star\right)\right)\right\|_F + 2\nu\eta^2 \mathbb{E}_{\mathcal{B}_t} \left\|\mathcal{P}_\Omega\left(\nabla f_{\mathcal{B}_t}\left(\mathbf{\Theta}^\star\right)\right)\right\|_F^2$$

where the second line in the statistical error in SGD. With the definition of $\kappa_1, \sigma_{GD}^2$,

$$\mathbb{E}_{\mathcal{B}_t}\left[\left\|\widetilde{\mathbf{\Theta}}^t - \mathbf{\Theta}^\star\right\|_F^2\right] \leq \kappa_1 \left\|\mathbf{\Theta}^{t-1} - \mathbf{\Theta}^\star\right\|_F^2 + \sigma_{GD}^2$$

Now we turn to $\mathbf{\Theta}^t$ which is given by

$$\mathbf{\Theta}^t = \mathcal{H}_k\left(\mathbf{\Theta}^{t-1} - \eta \mathcal{P}_{\widetilde{\Lambda}\cup\mathrm{supp}(\mathbf{\Theta}^{t-1})}\left(\mathbf{G}_t\right)\right)$$

It is very similar to $\widetilde{\mathbf{\Theta}}^t$, except that $\Lambda_{2k}$ is replaced by $\widetilde{\Lambda}$, which is the support we actually obtain. By definition, we have either $\Lambda_\Delta \subseteq \Lambda_{2k}$ or $\Lambda_{2k} \subseteq \Lambda_\Delta$. Recall that $\Lambda = \Lambda_{2k} \cap \Lambda_\Delta$ and in this case where ATEE recovers $\Lambda$, we have $\Lambda \subseteq \widetilde{\Lambda}$. Thus it is either $\Lambda_{2k} \subseteq \widetilde{\Lambda}$ or $\Lambda_\Delta \subseteq \widetilde{\Lambda}$.

1. $\Lambda_{2k} \subseteq \widetilde{\Lambda}$. In this case, simply applying corollary 12 with $G = \mathcal{P}_{\widetilde{\Lambda}\cup\mathrm{supp}(\mathbf{\Theta}^{t-1})}\left(\mathbf{G}_t\right)$, we have

   $$\mathbf{\Theta}^t = \mathcal{H}_k\left(\mathbf{\Theta}^{t-1} - \eta\mathcal{P}_{\widetilde{\Lambda}\cup\mathrm{supp}(\mathbf{\Theta}^{t-1})}\left(\mathbf{G}_t\right)\right) = \mathcal{H}_k\left(\mathbf{\Theta}^{t-1} - \eta\mathcal{P}_{\Lambda_{2k}\cup\mathrm{supp}(\mathbf{\Theta}^{t-1})}\left(\mathbf{G}_t\right)\right) = \widetilde{\mathbf{\Theta}}^t$$

   Also, by $\Lambda_{2k} \subseteq \Lambda_\Delta$, we know that $k_\Delta = |\Lambda_{2k}\backslash\Lambda_\Delta| = 0$, which indicates $\sigma_{\Delta|GD}^2 = 0$.

2. $\Lambda_\Delta \subseteq \widetilde{\Lambda}$. Here we can apply Corollary 13 and have,

   $$\|\mathbf{\Theta}^t - \widetilde{\mathbf{\Theta}}^t\|_F \leq 2\eta\Delta\sqrt{k_\Delta}, \quad \|\mathbf{\Theta}^t - \widetilde{\mathbf{\Theta}}^t\|_F^2 \leq 2\eta^2\Delta^2 k_\Delta$$

   Thus, we can bound the error $\mathbb{E}_{\mathcal{B}_t}\left[\|\mathbf{\Theta}^t - \mathbf{\Theta}^\star\|_F^2\right]$ as,

   $$\begin{aligned}
   \mathbb{E}_{\mathcal{B}_t}\left[\|\mathbf{\Theta}^t - \mathbf{\Theta}^\star\|_F^2\right] &= \mathbb{E}_{\mathcal{B}_t}\left[\left\|\widetilde{\mathbf{\Theta}}^t - \mathbf{\Theta}^\star\right\|_F^2\right] + 2\mathbb{E}_{\mathcal{B}_t}\left[\left\langle\widetilde{\mathbf{\Theta}}^t - \mathbf{\Theta}^\star, \mathbf{\Theta}^t - \widetilde{\mathbf{\Theta}}^t\right\rangle\right] + \mathbb{E}_{\mathcal{B}_t}\left[\|\mathbf{\Theta}^t - \widetilde{\mathbf{\Theta}}^t\|_F^2\right] \\
   &\leq \mathbb{E}_{\mathcal{B}_t}\left[\left\|\widetilde{\mathbf{\Theta}}^t - \mathbf{\Theta}^\star\right\|_F^2\right] + 4\sqrt{k_\Delta}\eta\sqrt{k}\omega\Delta + 2k_\Delta\eta^2\Delta^2 \\
   &= \mathbb{E}_{\mathcal{B}_t}\left[\left\|\widetilde{\mathbf{\Theta}}^t - \mathbf{\Theta}^\star\right\|_F^2\right] + \sigma_{\Delta|GD}^2
   \end{aligned}$$

Combining the two cases above, we have the desired convergence rate for $\Lambda \subseteq \widetilde{\Lambda}$.

**ATEE fails:** $\Lambda \not\subset \widetilde{\Lambda}$. This is the worst case when support recovery completely fails and we have no control over $\widetilde{\Lambda}$. The update in this case is

$$\mathbf{\Theta}^t = \mathcal{H}_k\left(\widetilde{\mathbf{Z}}^t\right), \quad \mathbf{Z}^t = \mathbf{\Theta}^{t-1} - \eta\left[\mathbf{G}^t - \mathcal{P}_{\widetilde{\Lambda}^C\backslash\mathrm{supp}(\mathbf{\Theta}^{t-1})}(\mathbf{G}^t)\right]$$

Similar as the previous case, let $\Omega = \mathrm{supp}(\mathbf{\Theta}^{t-1}) \cup \mathrm{supp}(\mathbf{\Theta}^t) \cup \mathrm{supp}(\mathbf{\Theta}^\star)$, we have

$$\left\|\mathbf{\Theta}^t - \mathbf{\Theta}^\star\right\|_F^2 \leq \nu\left\|\mathcal{P}_\Omega\left(\mathbf{Z}^t\right) - \mathbf{\Theta}^\star\right\|_F^2$$

and

$$\begin{aligned}
\left\|\mathcal{P}_\Omega\left(\mathbf{Z}^t\right) - \mathbf{\Theta}^\star\right\|_F^2 &\leq \left\|\mathbf{\Theta}^{t-1} - \mathbf{\Theta}^\star\right\|_F^2 + \eta^2\left\|\mathcal{P}_\Omega\left(\mathbf{G}^t - \mathcal{P}_{\widetilde{\Lambda}^C\backslash\mathrm{supp}(\mathbf{\Theta}^{t-1})}(\mathbf{G}^t)\right)\right\|_F^2 \\
&\quad - 2\eta\left\langle\mathbf{\Theta}^{t-1} - \mathbf{\Theta}^\star, \mathcal{P}_\Omega(\mathbf{G}^t)\right\rangle \\
&\quad + 2\eta\left\langle\mathbf{\Theta}^{t-1} - \mathbf{\Theta}^\star, \mathcal{P}_\Omega\mathcal{P}_{\widetilde{\Lambda}\backslash\mathrm{supp}(\mathbf{\Theta}^{t-1})}(\mathbf{G}^t)\right\rangle \\
&\leq \left\|\mathbf{\Theta}^{t-1} - \mathbf{\Theta}^\star\right\|_F^2 + \eta^2\left\|\mathcal{P}_\Omega\left(\mathbf{G}^t\right)\right\|_F^2 - 2\eta\left\langle\mathbf{\Theta}^{t-1} - \mathbf{\Theta}^\star, \mathcal{P}_\Omega(\mathbf{G}^t)\right\rangle \\
&\quad + 2\eta\left\langle\mathbf{\Theta}^{t-1} - \mathbf{\Theta}^\star, \mathcal{P}_\Omega\mathcal{P}_{\widetilde{\Lambda}\backslash\mathrm{supp}(\mathbf{\Theta}^{t-1})}(\mathbf{G}^t)\right\rangle
\end{aligned}$$

The bound for the first three terms are same as the bound for Equation (5). It left to bound the last term,

$$\left\langle \Theta^{t-1} - \Theta^\star, \mathcal{P}_\Omega \mathcal{P}_{\widetilde{\Lambda} \setminus \mathrm{supp}(\Theta^{t-1})}(\mathbf{G}^t) \right\rangle$$

$$\leq \left\| \Theta^{t-1} - \Theta^\star \right\|_F \left\| \mathcal{P}_\Omega \mathcal{P}_{\widetilde{\Lambda} \setminus \mathrm{supp}(\Theta^{t-1})} \left( \nabla f_{\mathcal{B}_t}(\Theta^{t-1}) \right) \right\|_F$$

$$\leq \left\| \Theta^{t-1} - \Theta^\star \right\|_F \left\| \mathcal{P}_\Omega \left( \nabla f_{\mathcal{B}_t}(\Theta^{t-1}) - \nabla f_{\mathcal{B}_t}(\Theta^*) \right) \right\|_F + \left\| \Theta^{t-1} - \Theta^\star \right\|_F \left\| \mathcal{P}_\Omega \left( \nabla f_{\mathcal{B}_t}(\Theta^*) \right) \right\|_F$$

$$\leq \left\| \Theta^{t-1} - \Theta^\star \right\|_F \left\| \nabla f_{\mathcal{B}_t}(\Theta^{t-1}) - \nabla f_{\mathcal{B}_t}(\Theta^*) \right\|_F + \left\| \Theta^{t-1} - \Theta^\star \right\|_F \left\| \mathcal{P}_\Omega \left( \nabla f_{\mathcal{B}_t}(\Theta^*) \right) \right\|_F$$

$$\leq L_{2k} \left\| \Theta^{t-1} - \Theta^\star \right\|_F^2 + 2\sqrt{k}\omega \left\| \mathcal{P}_\Omega \left( \nabla f_{\mathcal{B}_t}(\Theta^*) \right) \right\|_F$$

Putting the bounds together, we have

$$\mathbb{E}_{\mathcal{B}_t} \left[ \left\| \Theta^t - \Theta^\star \right\|_F^2 \right] \leq \nu \left( 1 - 2\eta\alpha_{2k} + 2\eta^2 L_{2k}^2 + 2\eta L_{2k} \right) \left\| \Theta^{t-1} - \Theta^\star \right\|_F^2$$
$$+ 4\nu\eta\sqrt{k}\omega \left\| \mathcal{P}_\Omega \left( \nabla F(\Theta^\star) \right) \right\|_F + 2\nu\eta^2 \mathbb{E}_{\mathcal{B}_t} \left\| \mathcal{P}_\Omega \left( \nabla f_{\mathcal{B}_t}(\Theta^\star) \right) \right\|_F^2$$
$$+ 4\nu\eta\sqrt{k}\omega \mathbb{E}_{\mathcal{B}_t} \left\| \mathcal{P}_\Omega \left( \nabla f_{\mathcal{B}_t}(\Theta^\star) \right) \right\|_F$$

Define $\sigma_{Fail|GD}^2 = \max_{|\Omega| \leq 2k+K} \left[ 4\nu\eta\sqrt{k}\omega \mathbb{E}_{\mathcal{B}_t} \left[ \left\| \mathcal{P}_\Omega \left( \nabla f_{\mathcal{B}_t}(\Theta^\star) \right) \right\|_F \right] \right]$ Then,

$$\mathbb{E}_{\mathcal{B}_t} \left[ \left\| \Theta^t - \Theta^\star \right\|_F^2 \right] \leq \nu \left( 1 - 2\eta\alpha_{2k} + 2\eta^2 L_{2k}^2 + 2\eta L_{2k} \right) \left\| \Theta^{t-1} - \Theta^\star \right\|_F^2 + \sigma_{GD}^2 + \sigma_{Fail|GD}^2$$

∎

### E.4 Proof of Theorem 4

*Proof.* With the definition of $\sigma_1^2, \sigma_2^2, \kappa_1, \kappa_2$, the per-round convergence result of Theorem 3 can be rewritten as:

1. Success Case:

$$\mathbb{E}_{B_{t+1}} \left[ \left\| \Theta^{t+1} - \Theta^\star \right\|_F^2 + \frac{\sigma_1^2}{\kappa_1 - 1} \right] \leq \kappa_1 \mathbb{E}_{B_t} \left[ \left\| \Theta^t - \Theta^\star \right\|_F^2 + \frac{\sigma_1^2}{\kappa_1 - 1} \right]$$

2. Failure Case:

$$\mathbb{E}_{B_{t+1}} \left[ \left\| \Theta^{t+1} - \Theta^\star \right\|_F^2 + \frac{\sigma_1^2}{\kappa_1 - 1} \right] \leq \kappa_2 \mathbb{E}_{B_t} \left[ \left\| \Theta^t - \Theta^\star \right\|_F^2 + \frac{\sigma_1^2}{\kappa_1 - 1} \right] + (\kappa_2 - 1) \left( \frac{\sigma_2^2}{\kappa_2 - 1} - \frac{\sigma_1^2}{\kappa_1 - 1} \right)$$

For each iteration, the count sketch succeeds with probability $1 - \frac{1}{c}$. Denote the success indicator at iteration $t$ as $\phi_t$, and let $\Phi_t = \{\phi_0, \phi_1, ..., \phi_t\}$, we can combine those two cases and obtain,

$$\mathbb{E}_{B_{t+1}, \Phi_{t+1}} \left[ \left\| \Theta^{t+1} - \Theta^\star \right\|_F^2 + \frac{\sigma_1^2}{\kappa_1 - 1} \right] \leq \left( \kappa_1 + \frac{1}{c} (\kappa_1 - \kappa_2) \right) \mathbb{E}_{B_t, \Phi_t} \left[ \left\| \Theta^t - \Theta^\star \right\|_F^2 + \frac{\sigma_1^2}{\kappa_1 - 1} \right]$$
$$+ \frac{1}{c} (\kappa_2 - 1) \left( \frac{\sigma_2^2}{\kappa_2 - 1} - \frac{\sigma_1^2}{\kappa_1 - 1} \right)$$

With a telescope sum, we have the desired error bound. ∎

### E.5 Proof of Theorem 5

*Proof.* We first vectorize the quadratic features. For an arbitrary data point $\mathbf{x}, y$, we know that $\mathbf{x} \in \mathbb{R}^p$, where the first $p-1$ coordinates independently come from a zero mean, bounded distribution and the last coordinate is a constant 1. Denote the $i$-th coordinate of $\mathbf{x}$ as $x_i$, we first vectorize the quadratic features, define

$$\mathbf{v} = [x_1 x_2, x_1 x_3, ..., x_{p-1} x_p]^\top$$

Here, for the quadratic terms, we only consider the interaction terms with no squared terms like $x_i^2$. Given that there will only be $p$ different squared terms, one can regress with the squared terms first, and the residual model will have no dependency on the squared terms. Replace $x_p$ with 1, we have

$$\mathbf{v} = [x_1 x_2, x_1 x_3, ..., x_{p-2} x_{p-1}, x_1, ..., x_{p-1}]^\top$$

Given that $x_1, ..., x_{p-1}$ are all i.i.d. and zero mean, after normalizing the variance of $x_i$, it is not hard to verify that

$$\mathbb{E}[\mathbf{v}\mathbf{v}^\top] = \mathbf{I}$$

Given that $\alpha, L$ are the smallest and largest eigenvalue of $\mathbb{E}[\mathbf{v}\mathbf{v}^\top]$, asymptotically, we have the strong convexity and smoothness parameter $\alpha = L = 1$, which directly implies that the restricted version $\alpha_k = L_k = 1$. Thus the deterministic requirement can be easily satisfied with infinite sample. Now we turn to the minimum sample we need to have the desired $\alpha_k, L_k$.

To show the $k$-restricted strong convexity and smoothness, we will first focus on an arbitrary $k$ sub-matrix of $\mathbb{E}[\mathbf{v}\mathbf{v}^\top]$ and show the concentration. Then the desired claim will follow by applying a union bound over all $k$ sub-matrices.

Denote a set of indices $\mathcal{S} \subseteq \{1, ..., |\mathbf{v}|\}$, where $|\mathcal{S}| = k$. Define the corresponding sub-vector drawn from $\mathbf{v}$ as $\mathbf{z} = \mathbf{v}_\mathcal{S}$. Define the restricted expected Hessian matrix as $\mathbf{H} = \mathbb{E}[\mathbf{z}\mathbf{z}^\top]$, let the finite sample Hessian matrix as $\widehat{\mathbf{H}} = \frac{1}{m} \sum_{i=1}^m \mathbf{z}_i \mathbf{z}_i^\top$. Denote the difference as

$$\mathbf{D}_i = \frac{1}{m} \mathbf{z}_i \mathbf{z}_i^\top - \frac{1}{m} \mathbb{E}[\mathbf{z}\mathbf{z}^\top]$$

$$\mathbf{D} = \sum_{i=1}^m \mathbf{D}_i = \frac{1}{m} \sum_{i=1}^m \mathbf{z}_i \mathbf{z}_i^\top - \mathbb{E}[\mathbf{z}\mathbf{z}^\top] = \widehat{\mathbf{H}} - \mathbf{H}$$

Given that $\mathbf{H} = \mathbf{I}$, we can show the concentration of $\alpha_k, L_k$ as long as we can control $\mathbf{D}$. Given that $\mathbf{v}$ is bounded, we know that $\mathbf{v}$ is sub-gaussian. Thus, bounding $\|\mathbf{D}\|$ is equivalent to showing concentration of the covariance matrix estimation of sub-gaussian random vectors. Using the Corollary 5.50 in [30], we have that with $m \geq C(t/\epsilon)^2 k$, where $C$ depends only on the sub-gaussian norm $\|\mathbf{v}\|_{\psi_2}$. It's not hard to verify that $\|\mathbf{v}\|_{\psi_2} \leq B$. Then we have that

$$\mathbb{P}(\|\mathbf{D}\| \geq \epsilon) \leq 2 \exp(-t^2 k)$$

Thus we obtain the bound for one particular $k$-sub-matrix. Taking an union bound over all $k$-sub-matrices, we have that

$$\mathbb{P}(L_k \geq 1 + \epsilon) \leq \binom{|\mathbf{v}|}{k} \mathbb{P}(\|\mathbf{D}\| \geq \epsilon)$$
$$\leq \exp\left(k \log(2ep^2/k) - t^2 k\right)$$

By choosing $t^2 \geq \log(p)$, which implies that $m \gtrsim Bk \log(p)/\epsilon^2$, we have $L_k \leq 1 + \epsilon$ with high probability. With a symmetric argument, we know that under same condition, we have $\alpha_k \geq 1 - \epsilon$ with high probability.

∎

### E.6   Proof of Corollary 6

*Proof.* With Theorem 4 showing the linear convergence, we know that the per iteration complexity dominates the overall complexity of IntHT. By Theorem 1 and Lemma 2, we show setting $b$ to $\mathcal{O}(k)$ is sufficient for ATEE to recover the support. Theorem 5 provides that the minimum batch size $m$ required for quadratic regression is $\mathcal{O}(k \log p)$. Combining those results, we conclude that the complexity of IntHT is $\widetilde{\mathcal{O}}(k(k + p))$. In the regime when $k$ is $\mathcal{O}(p^\gamma)$ for $\gamma < 1$, IntHT achieves sub-quadratic complexity. ∎

# F  Proofs for Appendix C

## F.1  Proof of Theorem 7

*Proof.* Similar to the **ATEE succeeds** case, define

$$\widetilde{\boldsymbol{\Theta}}_{j+1}^i = \mathcal{H}_k\left(\boldsymbol{\Theta}_j^i - \eta \mathcal{P}_{\Lambda_{2k} \cup \text{supp}(\boldsymbol{\Theta}_j^i)}\left(\mathbf{G}_j^i\right)\right) = \mathcal{H}_k\left(\boldsymbol{\Theta}_j^i - \eta \mathbf{G}_j^i\right)$$

The second equality is given by Corollary 12. By applying Corollary 13, we can also have

$$\|\boldsymbol{\Theta}_{j+1}^i - \widetilde{\boldsymbol{\Theta}}_{j+1}^i\|_F \le 2\eta\Delta\sqrt{k_\Delta}, \quad \|\boldsymbol{\Theta}_{j+1}^i - \widetilde{\boldsymbol{\Theta}}_{j+1}^i\|_F^2 \le 2\eta^2\Delta^2 k_\Delta$$

which further implies

$$\begin{aligned}
\left\|\boldsymbol{\Theta}_{j+1}^i - \boldsymbol{\Theta}^\star\right\|_F^2 &= \left\|\boldsymbol{\Theta}_{j+1}^i - \widetilde{\boldsymbol{\Theta}}_{j+1}^i\right\|_F^2 + \left\|\widetilde{\boldsymbol{\Theta}}_{j+1}^i - \boldsymbol{\Theta}^\star\right\|_F^2 + 2\left\langle \boldsymbol{\Theta}_{j+1}^i - \widetilde{\boldsymbol{\Theta}}_{j+1}^i, \widetilde{\boldsymbol{\Theta}}_{j+1}^i - \boldsymbol{\Theta}^\star\right\rangle \\
&\le \left\|\widetilde{\boldsymbol{\Theta}}_{j+1}^i - \boldsymbol{\Theta}^\star\right\|_F^2 + 4\sqrt{k_\Delta}\eta\sqrt{k}\omega\Delta + 2k_\Delta\eta^2\Delta^2 \\
&= \left\|\widetilde{\boldsymbol{\Theta}}_{j+1}^i - \boldsymbol{\Theta}^\star\right\|_F^2 + \sigma_{\Delta|SVRG}^{\prime 2}
\end{aligned}$$

The last equality defines $\sigma_{\Delta|SVRG}^{\prime 2}$. To bound $\left\|\widetilde{\boldsymbol{\Theta}}_{j+1}^i - \boldsymbol{\Theta}^\star\right\|_F^2$, the high level idea is similar to the proof of Theorem 10 in [26]. We first define,

$$\mathbf{Z}_{j+1}^i = \boldsymbol{\Theta}_j^i - \eta\mathbf{G}_j^i$$

Let $\Omega = \text{supp}(\boldsymbol{\Theta}_j^i) \cup \text{supp}(\widetilde{\boldsymbol{\Theta}}_{j+1}^i) \cup \text{supp}(\boldsymbol{\Theta}^\star)$,

$$\left\|\widetilde{\boldsymbol{\Theta}}_{j+1}^i - \boldsymbol{\Theta}^\star\right\|_F^2 = \left\|\mathcal{H}_k\left(\mathbf{Z}_{j+1}^i\right) - \boldsymbol{\Theta}^\star\right\|_F^2 = \left\|\mathcal{H}_k\left(\mathcal{P}_\Omega\left(\mathbf{Z}_{j+1}^i\right)\right) - \boldsymbol{\Theta}^\star\right\|_F^2 \le \nu\left\|\mathcal{P}_\Omega\left(\mathbf{Z}_{j+1}^i\right) - \boldsymbol{\Theta}^\star\right\|_F^2$$

where the last inequality follows from Lemma 9. Thus we have

$$\begin{aligned}
&\mathbb{E}_{b_j}\left[\left\|\widetilde{\boldsymbol{\Theta}}_{j+1}^i - \boldsymbol{\Theta}^\star\right\|_F^2\right] \\
\le &\mathbb{E}_{b_j}\left[\nu\left\|\mathcal{P}_\Omega\left(\mathbf{Z}_{j+1}^i\right) - \boldsymbol{\Theta}^\star\right\|_F^2\right] \\
= &\mathbb{E}_{b_j}\left[\nu\left\|\boldsymbol{\Theta}_j^i - \boldsymbol{\Theta}^\star\right\|_F^2 + \nu\eta^2\left\|\mathcal{P}_\Omega(\mathbf{G}_j^i)\right\|_F^2 - 2\nu\eta\left\langle\boldsymbol{\Theta}_j^i - \boldsymbol{\Theta}^\star, \mathbf{G}_j^i\right\rangle\right]
\end{aligned}$$

The second term can be bounded by using Corollary 11 and we can take expectation directly on the third term, since $\mathbb{E}_{b_j}[\mathbf{G}_j^i] = \nabla F(\boldsymbol{\Theta}_j^i)$. For brevity, denote $L = L_{|\Omega|}$. We then have,

$$\begin{aligned}
&\mathbb{E}_{b_j}\left[\left\|\widetilde{\boldsymbol{\Theta}}_{j+1}^i - \boldsymbol{\Theta}^\star\right\|_F^2\right] \\
\le &\nu\left\|\boldsymbol{\Theta}_j^i - \boldsymbol{\Theta}^\star\right\|_F^2 + 4\nu\eta^2 L_{2k}\left[F(\boldsymbol{\Theta}_j^i) - F(\boldsymbol{\Theta}^\star) + F(\boldsymbol{\Theta}_0^i) - F(\boldsymbol{\Theta}^\star)\right] - 2\nu\eta\left\langle\boldsymbol{\Theta}_j^i - \boldsymbol{\Theta}^\star, \nabla F\left(\boldsymbol{\Theta}_j^i\right)\right\rangle \\
&- 4\nu\eta^2 L_{2k}\left\langle\nabla F(\boldsymbol{\Theta}^\star), \boldsymbol{\Theta}_j^i + \boldsymbol{\Theta}_0^i - 2\boldsymbol{\Theta}^\star\right\rangle + 4\nu\eta^2\left\|\mathcal{P}_\Omega\left(\nabla F\left(\boldsymbol{\Theta}^\star\right)\right)\right\|_F^2 \\
\le &\nu(1 - \eta\alpha_{2k})\left\|\boldsymbol{\Theta}_j^i - \boldsymbol{\Theta}^\star\right\|_F^2 - 2\nu\eta(1 - 2\eta L_{2k})\left[F(\boldsymbol{\Theta}_j^i) - F(\boldsymbol{\Theta}^\star)\right] + 4\nu\eta^2 L_{2k}\left[F(\boldsymbol{\Theta}_0^i) - F(\boldsymbol{\Theta}^\star)\right] \\
&+ 4\nu\eta^2 L_{2k}\left\|\nabla F(\boldsymbol{\Theta}^\star)\right\|_F\left\|\boldsymbol{\Theta}_j^i + \boldsymbol{\Theta}_0^i - 2\boldsymbol{\Theta}^\star\right\|_F + 4\nu\eta^2\left\|\mathcal{P}_\Omega\left(\nabla F\left(\boldsymbol{\Theta}^\star\right)\right)\right\|_F^2
\end{aligned}$$

where the first inequality plugs in the result from Corollary 11 and takes expectation of $\mathbf{G}_j^i$. The second inequality uses RSC property and Cauchy-Shwartz inequality. For brevity, define $\sigma' = \max_{|\Omega|=3k+K}\left\|\mathcal{P}_\Omega F(\boldsymbol{\Theta}^*)\right\|_F$, we have that

$$\begin{aligned}
\mathbb{E}_{b_j}\left[\left\|\widetilde{\boldsymbol{\Theta}}_{j+1}^i - \boldsymbol{\Theta}^\star\right\|_F^2\right] \le &\nu(1 - \eta\alpha_{2k})\left\|\boldsymbol{\Theta}_j^i - \boldsymbol{\Theta}^\star\right\|_F^2 - 2\nu\eta(1 - 2\eta L_{2k})\left[F(\boldsymbol{\Theta}_j^i) - F(\boldsymbol{\Theta}^\star))\right] \\
&+ 4\nu\eta^2 L_{2k}\left[F(\boldsymbol{\Theta}_0^i) - F(\boldsymbol{\Theta}^\star)\right] + 4\nu\eta\sigma'(4L_{2k}\sqrt{k}\omega + \sigma')
\end{aligned}$$

Thus for $\mathbb{E}_{b_j}\left[\left\|\mathbf{\Theta}_{j+1}^i - \mathbf{\Theta}^\star\right\|_F^2\right]$, we have

$$
\begin{aligned}
\mathbb{E}_{b_j}\left[\left\|\mathbf{\Theta}_{j+1}^i - \mathbf{\Theta}^\star\right\|_F^2\right] \leq & \nu(1 - \eta\alpha_{2k})\left\|\mathbf{\Theta}_j^i - \mathbf{\Theta}^\star\right\|_F^2 - 2\nu\eta(1 - 2\eta L_{2k})\left[F(\mathbf{\Theta}_j^i) - F(\mathbf{\Theta}^\star))\right] \\
& + 4\nu\eta^2 L_{2k}\left[F(\mathbf{\Theta}_0^i) - F(\mathbf{\Theta}^\star)\right] + 4\nu\eta\sigma'(4L_{2k}\sqrt{k}\omega + \sigma') + \sigma_{\Delta|SVRG}^2
\end{aligned}
$$

By a telescope sum, define $B_t = \{b_1, b_2, ..., b_t\}$

$$
\begin{aligned}
\mathbb{E}_{B_t}\left[\left\|\mathbf{\Theta}_t^i - \mathbf{\Theta}^\star\right\|_F^2\right] \leq & [\nu(1 - \eta\alpha_{2k}) - 1]\sum_{j=0}^{t-1}\left\|\mathbf{\Theta}_{j+1}^i - \mathbf{\Theta}^\star\right\|_F^2 + \left\|\mathbf{\Theta}_0^i - \mathbf{\Theta}^\star\right\|_F \\
& - 2\nu\eta(1 - 2\eta L_{2k})\sum_{j=0}^{t-1}\left[F(\mathbf{\Theta}_{j+1}^i) - F(\mathbf{\Theta}^\star))\right] \\
& + 4\nu\eta^2 L_{2k}m\left[F(\mathbf{\Theta}_0^i) - F(\mathbf{\Theta}^\star)\right] + 4\nu\eta\sigma'(4L_{2k}\sqrt{k}\omega + \sigma')m + m\sigma_{\Delta|SVRG}^2 \\
= & [\nu(1 - \eta\alpha_{2k}) - 1]m\mathbb{E}_{B_t,j'}\left\|\mathbf{\Theta}_0^{i+1} - \mathbf{\Theta}^\star\right\|_F^2 + \left\|\mathbf{\Theta}_0^i - \mathbf{\Theta}^\star\right\|_F \\
& - 2\nu\eta(1 - 2\eta L_{2k})\mathbb{E}_{B_t,j'}\left[F(\mathbf{\Theta}_0^{i+1}) - F(\mathbf{\Theta}^\star))\right] \\
& + 4\nu\eta^2 L_{2k}m\left[F(\mathbf{\Theta}_0^i) - F(\mathbf{\Theta}^\star)\right] + 4\nu\eta\sigma'(4L_{2k}\sqrt{k}\omega + \sigma')m + m\sigma_{\Delta|SVRG}^2
\end{aligned}
$$

By using RSC, we have $\frac{2}{\alpha_{2k}}\left[F(\mathbf{\Theta}_0^i) - F(\mathbf{\Theta}^\star) - \left\langle\nabla F(\mathbf{\Theta}^\star), \mathbf{\Theta}_0^i - \mathbf{\Theta}^\star\right\rangle\right] \geq \left\|\mathbf{\Theta}_0^i - \mathbf{\Theta}^\star\right\|_F^2$, thus

$$
\begin{aligned}
\mathbb{E}_{B_t}\left[\left\|\mathbf{\Theta}_t^i - \mathbf{\Theta}^\star\right\|_F^2\right] \leq & [\nu(1 - \eta\alpha_{2k}) - 1]m\mathbb{E}_{B_t,j'}\left\|\mathbf{\Theta}_0^{i+1} - \mathbf{\Theta}^\star\right\|_F^2 \\
& - 2\nu\eta(1 - 2\eta L_{2k})\mathbb{E}_{B_t,j'}\left[F(\mathbf{\Theta}_0^{i+1}) - F(\mathbf{\Theta}^\star))\right] \\
& + \left(\frac{2}{\alpha_{2k}} + 4\nu\eta^2 L_{2k}m\right)\left[F(\mathbf{\Theta}_0^i) - F(\mathbf{\Theta}^\star)\right] \\
& + 4\nu\eta\sigma'(4L_{2k}\sqrt{k}\omega + \sigma')m + \frac{2}{\alpha_{2k}}\sqrt{k}\omega\sigma' + m\sigma_{\Delta|SVRG}^2
\end{aligned}
$$

By assumption, we have $[\nu(1 - \eta\alpha_{2k}) - 1] \leq 0$. For simplicity, define

$$
\sigma_{SVRG} = \frac{4\nu\eta\sigma'(4L_{2k}\sqrt{k}\omega + \sigma')m + \frac{2}{\alpha_{2k}}\sqrt{k}\omega\sigma' + m\sigma_{\Delta|SVRG}^2}{2\nu\eta(1 - 2\eta L_{2k})m}
$$

Choosing $\eta < \frac{1}{2L_{2k}}$, we have

$$
\mathbb{E}_{B_t}\left[F(\mathbf{\Theta}_0^{i+1}) - F(\mathbf{\Theta}^\star))\right] \leq \left(\frac{1}{\alpha_{2k}\nu\eta(1 - 2\eta L_{2k})m} + \frac{2\eta L_{2k}}{1 - 2\eta L_{2k}}\right)\left[F(\mathbf{\Theta}_0^i) - F(\mathbf{\Theta}^\star)\right] + \sigma_{SVRG}
$$

Typically $m$ is quite large, thus for the condition of convergence, we require that

$$
\frac{2\eta L_{2k}}{1 - 2\eta L_{2k}} < 1 \Rightarrow \eta < \frac{1}{4L_{2k}}
$$

Define,

$$
\kappa_{SVRG} = \frac{1}{\alpha_{2k}\nu\eta(1 - 2\eta L_{2k})m} + \frac{2\eta L_{2k}}{1 - 2\eta L_{2k}}
$$

We have the linear convergence given by

$$
\mathbb{E}_{B_t}\left[F(\mathbf{\Theta}_0^{i+1}) - F(\mathbf{\Theta}^\star))\right] \leq \kappa_{SVRG}\left[F(\mathbf{\Theta}_0^i) - F(\mathbf{\Theta}^\star)\right] + \sigma_{SVRG}
$$

$\blacksquare$

# G Proofs for Appendix D

## G.1 Proof of Lemma 9

*Proof.* We give the proof here for completeness. Also, the proof here is much concise than the original proof in [26] and a related result shown in [14].

$$\|\mathcal{H}_k(\mathbf{B}) - \mathbf{\Theta}\|_F^2 = \underbrace{\left\|\mathcal{P}_{\mathrm{supp}(\mathcal{H}_k(\mathbf{B}))\backslash\mathrm{supp}(\mathbf{\Theta})}(\mathbf{B})\right\|_F^2}_{\|\mathbf{B}_1\|_F^2} + \underbrace{\left\|\mathcal{P}_{\mathrm{supp}(\mathcal{H}_k(\mathbf{B}))\cap\mathrm{supp}(\mathbf{\Theta})}(\mathbf{B}-\mathbf{\Theta})\right\|_F^2}_{\|\mathbf{B}_2-\mathbf{\Theta}_2\|_F^2}$$

$$+ \underbrace{\left\|\mathcal{P}_{\mathrm{supp}^c(\mathcal{H}_k(\mathbf{B}))\cap\mathrm{supp}(\mathbf{\Theta})}(\mathbf{\Theta})\right\|_F^2}_{\|\mathbf{\Theta}_3\|_F^2}$$

On the other hand,

$$\|\mathbf{B} - \mathbf{\Theta}\|_F^2 = \underbrace{\left\|\mathcal{P}_{\mathrm{supp}(\mathcal{H}_k(\mathbf{B}))\backslash\mathrm{supp}(\mathbf{\Theta})}(\mathbf{B})\right\|_F^2}_{\|\mathbf{B}_1\|_F^2} + \underbrace{\left\|\mathcal{P}_{\mathrm{supp}(\mathcal{H}_k(\mathbf{B}))\cap\mathrm{supp}(\mathbf{\Theta})}(\mathbf{B}-\mathbf{\Theta})\right\|_F^2}_{\|\mathbf{B}_2-\mathbf{\Theta}_2\|_F^2}$$

$$+ \underbrace{\left\|\mathcal{P}_{\mathrm{supp}^c(\mathcal{H}_k(\mathbf{B}))\cap\mathrm{supp}(\mathbf{\Theta})}(\mathbf{B}-\mathbf{\Theta})\right\|_F^2}_{\|\mathbf{B}_3-\mathbf{\Theta}_3\|_F^2} + \underbrace{\left\|\mathcal{P}_{\mathrm{supp}^c(\mathcal{H}_k(\mathbf{B}))\backslash\mathrm{supp}(\mathbf{\Theta})}(\mathbf{B})\right\|_F^2}_{\|\mathbf{B}_4\|_F^2}$$

$$\max_{\mathbf{B},\mathbf{\Theta}} \frac{\|\mathcal{H}_k(\mathbf{B}) - \mathbf{\Theta}\|_F^2}{\|\mathbf{B} - \mathbf{\Theta}\|_F^2}$$

$$= \max_{\mathbf{B},\mathbf{\Theta}} \frac{\|\mathbf{B}_1\|_F^2 + \|\mathbf{B}_2 - \mathbf{\Theta}_2\|_F^2 + \|\mathbf{\Theta}_3\|_F^2}{\|\mathbf{B}_1\|_F^2 + \|\mathbf{B}_2 - \mathbf{\Theta}_2\|_F^2 + \|\mathbf{B}_3 - \mathbf{\Theta}_3\|_F^2 + \|\mathbf{B}_4\|_F^2}$$

$$\leq \max_{\mathbf{B},\mathbf{\Theta}} \frac{\|\mathbf{B}_1\|_F^2 + \|\mathbf{B}_2 - \mathbf{\Theta}_2\|_F^2 + \|\mathbf{\Theta}_3\|_F^2}{\|\mathbf{B}_1\|_F^2 + \|\mathbf{B}_2 - \mathbf{\Theta}_2\|_F^2 + \|\mathbf{\Theta}_3\|_F^2 + \|\mathbf{B}_3\|_F^2 - 2\langle\mathbf{B}_3, \mathbf{\Theta}_3\rangle}$$

$$\leq \max\left\{1, \max_{\mathbf{B},\mathbf{\Theta}} \frac{|\mathrm{supp}(\mathbf{B}_1)|\,\mathbf{B}_{1,\min}{}^2 + \|\mathbf{\Theta}_3\|_F^2}{|\mathrm{supp}(\mathbf{B}_1)|\,\mathbf{B}_{1,\min}{}^2 + \|\mathbf{\Theta}_3\|_F^2 + \|\mathbf{B}_3\|_F^2 - 2\langle\mathbf{B}_3, \mathbf{\Theta}_3\rangle}\right\}$$

$$\leq \max\left\{1, \max_{\mathbf{B},\mathbf{\Theta}} \frac{|\mathrm{supp}(\mathbf{B}_1)|\,\mathbf{B}_{1,\min}{}^2 + \|\mathbf{\Theta}_3\|_F^2}{|\mathrm{supp}(\mathbf{B}_1)|\,\mathbf{B}_{1,\min}{}^2 + \|\mathbf{\Theta}_3\|_F^2 + |\mathrm{supp}(\mathbf{\Theta}_3)|\,\mathbf{B}_{1,\min}^2 - 2\mathbf{B}_{1,\min}\|\mathbf{\Theta}_3\|_1}\right\} \triangleq \gamma$$

We determine $\gamma$ by observing

$$\frac{|\mathrm{supp}(\mathbf{B}_1)|\,\mathbf{B}_{1,\min}{}^2 + \|\mathbf{\Theta}_3\|_F^2}{|\mathrm{supp}(\mathbf{B}_1)|\,\mathbf{B}_{1,\min}{}^2 + \|\mathbf{\Theta}_3\|_F^2 + |\mathrm{supp}(\mathbf{\Theta}_3)|\,\mathbf{B}_{1,\min}^2 - 2\mathbf{B}_{1,\min}\|\mathbf{\Theta}_3\|_1} \leq \gamma, \qquad \forall \mathbf{B}, \mathbf{\Theta}$$

$$\Leftrightarrow (\gamma-1)\|\mathbf{\Theta}_3\|_F^2 - 2\gamma\mathbf{B}_{1,\min}^2 + (\gamma|\mathrm{supp}(\mathbf{\Theta}_3)| + (\gamma-1)|\mathrm{supp}(\mathbf{B}_1)|)\mathbf{B}_{1,\min}^2 \geq 0, \qquad \forall \mathbf{B}, \mathbf{\Theta}$$

$$\Leftrightarrow 4\gamma^2\mathbf{B}_{1,\min}^2 \leq 4(\gamma-1)\left[\gamma + (\gamma-1)\frac{\mathrm{supp}(\mathbf{B}_1)}{\mathrm{supp}(\mathbf{\Theta}_3)}\right]\mathbf{B}_{1,\min}^2$$

$$\Leftarrow \gamma^2 \leq (\gamma-1)\left[\gamma + (\gamma-1)\frac{1}{\rho}\right]$$

$$\Leftarrow \gamma = 1 + \frac{\rho + \sqrt{(4+\rho)\rho}}{2}, \qquad \text{where } \rho = \frac{\min\{K, p^2 - k\}}{k - K + \min\{K, p^2 - k\}}$$

∎

## G.2 Proof of Corollary 10

*Proof.* Define the auxiliary function

$$g(\mathbf{\Xi}) := f(\mathbf{\Xi}) - \langle \nabla f(\mathbf{\Theta}), \mathbf{\Xi}\rangle \tag{6}$$

Notice that the gradient of $g(\cdot)$ satisfies:

$$\|\nabla g(\boldsymbol{\Xi}) - \nabla g(\boldsymbol{\Xi}')\|_F = \|\nabla f(\boldsymbol{\Xi}) - \nabla f(\boldsymbol{\Xi}')\|_F \leq L_{\|\boldsymbol{\Xi}-\boldsymbol{\Xi}'\|_0} \|\boldsymbol{\Xi} - \boldsymbol{\Xi}'\|_F$$

which implies

$$g(\boldsymbol{\Xi}) - g(\boldsymbol{\Xi}') - \langle \nabla g(\boldsymbol{\Xi}'), \boldsymbol{\Xi} - \boldsymbol{\Xi}' \rangle \leq \frac{L_r}{2} \|\boldsymbol{\Xi} - \boldsymbol{\Xi}'\|_F^2$$

where $r = |\mathrm{supp}(\boldsymbol{\Xi} - \boldsymbol{\Xi}')|$. On the other hand,

$$g(\boldsymbol{\Xi}) - g(\boldsymbol{\Theta}) = f(\boldsymbol{\Xi}) - f(\boldsymbol{\Theta}) - \langle \nabla f(\boldsymbol{\Theta}), \boldsymbol{\Xi} - \boldsymbol{\Theta} \rangle \geq 0$$

as long as $f(\cdot)$ satisfies $|\mathrm{supp}(\boldsymbol{\Xi}) \cup \mathrm{supp}(\boldsymbol{\Theta})|$-RC. Take $\boldsymbol{\Xi} = \boldsymbol{\Theta}' - \frac{1}{L_{|\Omega|}} \mathcal{P}_\Omega \nabla g(\boldsymbol{\Theta}')$, $\boldsymbol{\Xi}' = \boldsymbol{\Theta}'$, then,

$$
\begin{aligned}
g(\boldsymbol{\Theta}) \leq & g(\boldsymbol{\Theta}' - \frac{1}{L_{|\Omega|}} \mathcal{P}_\Omega \nabla g(\boldsymbol{\Theta}')) \\
\leq & g(\boldsymbol{\Theta}') + \left\langle \nabla g(\boldsymbol{\Theta}'), -\frac{1}{L_{|\Omega|}} \mathcal{P}_\Omega \nabla g(\boldsymbol{\Theta}') \right\rangle + \frac{1}{2L_{|\Omega|}} \|\mathcal{P}_\Omega \nabla g(\boldsymbol{\Theta}')\|_F^2 \\
= & g(\boldsymbol{\Theta}') - \frac{1}{2L_{|\Omega|}} \|\mathcal{P}_\Omega \nabla g(\boldsymbol{\Theta}')\|_F^2
\end{aligned}
$$

Plug in the definition in Equation (6) gives the result we want. ∎

### G.3   Proof of Corollary 11

*Proof.*

1. SGD:

$$
\begin{aligned}
\mathbb{E}_{\mathcal{B}_t} \left\| \mathcal{P}_\Omega \left( \mathbf{G}^t \right) \right\|_F^2 = & \mathbb{E}_{\mathcal{B}_t} \left\| \mathcal{P}_\Omega \left( \nabla f_{\mathcal{B}_t} \left( \boldsymbol{\Theta}^{t-1} \right) \right) \right\|_F^2 \\
\leq & 2\mathbb{E}_{\mathcal{B}_t} \left\| \mathcal{P}_\Omega \left( \nabla f_{\mathcal{B}_t} \left( \boldsymbol{\Theta}^{t-1} \right) - \nabla f_{\mathcal{B}_t} \left( \boldsymbol{\Theta}^\star \right) \right) \right\|_F^2 + 2 \left\| \mathcal{P}_\Omega \left( \nabla f_{\mathcal{B}_t} \left( \boldsymbol{\Theta}^\star \right) \right) \right\|_F^2 \\
\leq & 2L_{2k}^2 \left\| \boldsymbol{\Theta}^{t-1} - \boldsymbol{\Theta}^\star \right\|_F^2 + 2 \left\| \mathcal{P}_\Omega \left( \nabla f_{\mathcal{B}_t} (\boldsymbol{\Theta}^\star) \right) \right\|_F^2
\end{aligned}
$$

The first inequality is by algebra, the second inequality holds by RSM.

2. SVRG:

$$
\begin{aligned}
\left\| \mathcal{P}_\Omega(\mathbf{G}_j^i) \right\|_F^2 = & \left\| \mathcal{P}_\Omega \left( \nabla f_{b_j} \left( \boldsymbol{\Theta}_j^i \right) - \nabla f_{b_j} \left( \boldsymbol{\Theta}_0^i \right) + \nabla F \left( \boldsymbol{\Theta}_0^i \right) \right) \right\|_F^2 \\
\leq & 2 \left\| \mathcal{P}_\Omega \left( \nabla f_{b_j} \left( \boldsymbol{\Theta}_j^i \right) - \nabla f_{b_j} \left( \boldsymbol{\Theta}^\star \right) \right) \right\|_F^2 \\
& + 2 \left\| \mathcal{P}_\Omega \left( \nabla f_{b_j} \left( \boldsymbol{\Theta}_0^i \right) - \nabla f_{b_j} \left( \boldsymbol{\Theta}^\star \right) - \nabla F \left( \boldsymbol{\Theta}_0^i \right) \right) \right\|_F^2
\end{aligned}
$$

Expand the later square, we have

$$
\begin{aligned}
\left\| \mathcal{P}_\Omega(\mathbf{G}_j^i) \right\|_F^2 \leq & 2 \left\| \mathcal{P}_\Omega \left( \nabla f_{b_j} \left( \boldsymbol{\Theta}_j^i \right) - \nabla f_{b_j} \left( \boldsymbol{\Theta}^\star \right) \right) \right\|_F^2 + 2 \left\| \mathcal{P}_\Omega \left( \nabla f_{b_j} \left( \boldsymbol{\Theta}_0^i \right) - \nabla f_{b_j} \left( \boldsymbol{\Theta}^\star \right) \right) \right\|_F^2 \\
& + 2 \left\| \mathcal{P}_\Omega \nabla F \left( \boldsymbol{\Theta}_0^i \right) \right\|_F^2 - 4 \left\langle \mathcal{P}_\Omega \left( \nabla f_{b_j} \left( \boldsymbol{\Theta}_0^i \right) - \nabla f_{b_j} \left( \boldsymbol{\Theta}^\star \right) \right), \mathcal{P}_\Omega \nabla F \left( \boldsymbol{\Theta}_0^i \right) \right\rangle
\end{aligned}
$$

By applying Corollary 10 to bound the first two terms, we have

$$
\begin{aligned}
\left\| \mathcal{P}_\Omega(\mathbf{G}_j^i) \right\|_F^2 \leq & 4L_{2k} \left[ f_{b_j}(\boldsymbol{\Theta}_j^i) - f_{b_j}(\boldsymbol{\Theta}^\star) - \langle \nabla f_{b_j}(\boldsymbol{\Theta}^\star), \boldsymbol{\Theta}_j^i - \boldsymbol{\Theta}^\star \rangle \right] \\
& + 4L_{2k} \left[ f_{b_j}(\boldsymbol{\Theta}_0^i) - f_{b_j}(\boldsymbol{\Theta}^\star) - \langle \nabla f_{b_j}(\boldsymbol{\Theta}^\star), \boldsymbol{\Theta}_0^i - \boldsymbol{\Theta}^\star \rangle \right] \\
& + 2 \left\| \mathcal{P}_\Omega \nabla F \left( \boldsymbol{\Theta}_0^i \right) \right\|_F^2 - 4 \left\langle \mathcal{P}_\Omega \left( \nabla f_{b_j} \left( \boldsymbol{\Theta}_0^i \right) - \nabla f_{b_j} \left( \boldsymbol{\Theta}^\star \right) \right), \mathcal{P}_\Omega \nabla F \left( \boldsymbol{\Theta}_0^i \right) \right\rangle
\end{aligned}
$$

Taking expectation over $b_j$, we have

$$
\begin{aligned}
\mathbb{E}_{b_j}\left\|\mathcal{P}_\Omega(\mathbf{G}_j^i)\right\|_F^2 \leq & 4L_{2k}\left[F(\mathbf{\Theta}_j^i) - F(\mathbf{\Theta}^\star)\right] + 4L_{2k}\left[F(\mathbf{\Theta}_0^i) - F(\mathbf{\Theta}^\star)\right] \\
& - 4L_{2k}\left\langle \nabla F(\mathbf{\Theta}^\star), \mathbf{\Theta}_j^i + \mathbf{\Theta}_0^j - 2\mathbf{\Theta}^\star\right\rangle \\
& + 2\left\langle 2\mathcal{P}_\Omega\left(\nabla F\left(\mathbf{\Theta}^\star\right)\right) - \mathcal{P}_\Omega\left(\nabla F\left(\mathbf{\Theta}_0^i\right)\right), \mathcal{P}_\Omega \nabla F\left(\mathbf{\Theta}_0^i\right)\right\rangle \\
= & 4L_{2k}\left[F(\mathbf{\Theta}_j^i) - F(\mathbf{\Theta}^\star)\right] + 4L_{2k}\left[F(\mathbf{\Theta}_0^i) - F(\mathbf{\Theta}^\star)\right] \\
& - 4L_{2k}\left\langle \nabla F(\mathbf{\Theta}^\star), \mathbf{\Theta}_j^i + \mathbf{\Theta}_0^j - 2\mathbf{\Theta}^\star\right\rangle \\
& + \left\|2\mathcal{P}_\Omega\left(\nabla F(\mathbf{\Theta}^\star)\right)\right\|_F^2 - \left\|2\mathcal{P}_\Omega(\nabla F(\mathbf{\Theta}^\star) - \nabla F(\mathbf{\Theta}_0^i))\right\|_F^2 \\
& - \left\|\mathcal{P}_\Omega\left(\nabla F(\mathbf{\Theta}_0^i)\right)\right\|_F^2 \\
\leq & 4L_{2k}\left[F(\mathbf{\Theta}_j^i) - F(\mathbf{\Theta}^\star)\right] + 4L_{2k}\left[F(\mathbf{\Theta}_0^i) - F(\mathbf{\Theta}^\star)\right] \\
& - 4L_{2k}\left\langle \nabla F(\mathbf{\Theta}^\star), \mathbf{\Theta}_j^i + \mathbf{\Theta}_0^j - 2\mathbf{\Theta}^\star\right\rangle + 4\left\|\mathcal{P}_\Omega\left(\nabla F(\mathbf{\Theta}^\star)\right)\right\|_F^2
\end{aligned}
$$

■

## G.4  Proof of Corollary 12

*Proof.* Denote $\mathbf{\Theta}^+ = \mathcal{H}_k\left(\mathbf{\Theta} - \eta\mathbf{G}\right)$. Define $\Lambda_{new}$ to be the indices set of $k$-largest elements in $G$ that doesn't belong to $\text{supp}(\mathbf{\Theta})$. It can be easily verified that

$$
\mathcal{H}_k\left(\mathbf{\Theta} - \eta\mathbf{G}\right) = \mathcal{H}_k\left(\mathbf{\Theta} - \eta\mathcal{P}_{\text{supp}(\mathbf{\Theta})\cup\Lambda_{new}}\left(\mathbf{G}\right)\right)
$$

Given that $|\text{supp}(\mathbf{\Theta})| \leq k$, by pigeonhole principle, we have $\Lambda_{new} \subseteq \Lambda_{2k}$, thus

$$
\mathcal{H}_k\left(\mathbf{\Theta} - \eta\mathcal{P}_{\text{supp}(\mathbf{\Theta})\cup\Lambda_{new}}\left(\mathbf{G}\right)\right) = \mathcal{H}_k\left(\mathbf{\Theta} - \eta\mathcal{P}_{\text{supp}(\mathbf{\Theta})\cup\Lambda_{2k}}\left(\mathbf{G}\right)\right)
$$

■

## G.5  Proof of Corollary 13

*Proof.* Define

$$
\Gamma_0 = \text{supp}(\mathbf{\Theta}^+) \cap \text{supp}(\widetilde{\mathbf{\Theta}}^+), \Gamma_1 = \text{supp}(\mathbf{\Theta}^+)\backslash\text{supp}(\widetilde{\mathbf{\Theta}}^+), \Gamma_2 = \text{supp}(\widetilde{\mathbf{\Theta}}^+)\backslash\text{supp}(\mathbf{\Theta}^+)
$$

We have that

$$
\mathbf{\Theta}^+ - \widetilde{\mathbf{\Theta}}^+ = \mathcal{P}_{\Gamma_1}(\mathbf{\Theta}^+) - \mathcal{P}_{\Gamma_2}(\widetilde{\mathbf{\Theta}}^+)
$$

By definition of $\mathbf{\Theta}^+, \widetilde{\mathbf{\Theta}}^+$, it is easy to verify that

$$
\|\mathcal{P}_{\Gamma_1}(\mathbf{\Theta}^+)\|_\infty \leq \|\mathcal{P}_{\Gamma_2}(\widetilde{\mathbf{\Theta}}^+)\|_\infty
$$

Also, since the elements that is greater than $\eta\Delta$ can only come from $\text{supp}(\mathbf{\Theta}) \cup \Lambda_\Delta$, and given that $\Lambda_\Delta \subseteq \widetilde{\Lambda}$, we know that

$$
i \in \text{supp}(\mathbf{\Theta}^+), \forall i \ s.t. \widetilde{\mathbf{\Theta}}_i^+ \geq \eta\Delta \Rightarrow i \in \Gamma_0, \forall i \ s.t. \widetilde{\mathbf{\Theta}}_i^+ \geq \eta\Delta
$$

Thus we have

$$
\|\mathcal{P}_{\Gamma_2}(\widetilde{\mathbf{\Theta}}^+)\|_\infty \leq \eta\Delta
$$

By Corollary 12, we know that

$$
\widetilde{\mathbf{\Theta}}^+ = \mathcal{H}_k\left(\mathbf{\Theta} - \eta\mathcal{P}_{\Lambda_{2k}\cup\text{supp}(\mathbf{\Theta})}(\mathbf{G}_t)\right)
$$

Thus we have $|\Gamma_2| \leq k_\Delta$ and given that $|\text{supp}(\mathbf{\Theta}^+)| = |\widetilde{\text{supp}}(\mathbf{\Theta}^+)|$, we have $|\Gamma_1| = |\Gamma_2|$. Thus,

$$
\left\|\mathbf{\Theta}^+ - \widetilde{\mathbf{\Theta}}^+\right\|_F = \left\|\mathcal{P}_{\Gamma_1}(\mathbf{\Theta}^+) - \mathcal{P}_{\Gamma_2}(\widetilde{\mathbf{\Theta}}^+)\right\|_F \leq \eta\Delta\sqrt{2k_\Delta}
$$

■