[Reviews · NeurIPS 2019]

Reviewer 1



This paper proposes a new algorithm – Interaction Hard Thresholding (IntHT) for the quadratic regression problem, a variant of Iterative Hard Thresholding that uses the special quadratic structure to devise a new way to (approx.) extract the top elements of a $p^2$ size gradient in sub-$p^2$ time and space. Then the authors show that IntHT linearly converges to a consistent estimate under standard high-dimensional sparse recovery assumptions. Perhaps, the algorithm is the first one to provably accurately solve quadratic regression problem in sub-quadratic time and space. Overall I find the results and the algorithms are quite interesting although the main idea of the new algorithm is quite simple: It mainly combines the existing technique from [20] with IHT to compute the top elements of the full gradient. The paper is well organized and presented. Minor comments: -As also noted in the paper, the quadratic regression problem has a strong background in real applications. Could you give a real example that $\Theta$ is sparse? - I am quite curious about the proof novelty. What is the key novel of the proof? ------After rebuttal----------------------- The authors have addressed my concerns well.

Reviewer 2



Originality: This paper applies approaches in Maximum Inner Product to the problem of quadratic regression in order to reduce the time and space complexities. It is also interesting that convergence analysis can be established for such methods. Quality: Although theoretical results have been established for the proposed methods, the strength and optimality of the established results are unclear. Furthermore, the numerical simulations do not show the trade-off between statistical and computational efficiencies. Clarity: The paper is basically well-written. Significance: The problem is not really well motivated, so it is hard to evaluate the significance of the proposed method. -----------After rebuttal---------- Given my previous concerns are mostly addressed, I changed my rating from 5 to 7.

Reviewer 3



In this paper, the authors study the problem of quadratic regression, whereby the response is modeled as function of both linear and quadratic terms, rather than the typical linear model. Naively, one would expect the dependence on the dimension of the problem to increase quadratically, and so the objective of this paper is to approximately capture the top entries of a (p^2-sized) gradient in time and space that are sub-p^2, under certain assumptions. Concretely, the authors propose the algorithm Interaction Hard Thresholding (IntHT), a variant of Iterative Hard Thresholding, which can recover a sparse solution (when such a solution is able to model the data well), with sub-quadratic complexity. Experiments done on synthetic data validate the theoretical conclusions. ====== Strengths ====== This work provides a new thresholding algorithm for the problem of quadratic regression that achieves sub-quadratic dependence on the dimension of the problem. It is also noted in the paper that the results do not follow immediately by previous analyses due to the difficulty of finding the exact top elements of the gradient. Thus, one of the key contributions of the paper is to show that linear convergence holds, even with only an approximate top element extraction. ====== Weaknesses ====== The main weakness of the paper is that it requires the assumption that there exists some sufficiently good solution that is also sparse. A discussion of how realistic this assumption is would be welcome, as when this assumption does not hold (i.e., only dense solutions suffice), sub-quadratic rates are no longer achieved. ====== Minor Comments ====== - under “Linear Model” on first page: extraneous “)” - p.7: obtian -> obtain - p.7: “despite the hard thresholding is not accurate?” -> “despite the hard thresholding not being accurate?” - p.7: different bs -> different b’s ========================================== The authors have addressed my concerns regarding the motivation for the setting.

[Author Response · NeurIPS 2019]

We thank all the reviewers for the helpful comments. Here, we address the main concerns raised by the reviewers.

**To Reviewer # 3** [1.1. Motivations for regression with sparse interaction terms.] Regression with interaction terms has been studied for a long time (see [1] and its citations). The main obstacle comes from the dramatically increased dimensionality due to considering the quadratic or even higher order interaction terms. An immediate remedy is adding constraints to the model, which helps to reduce the model complexity. In our work, we adopted the sparse constraint, for both computational efficiency and interpretability.

[1.2. Real application with sparse $\Theta$] The sparse interaction assumption also holds in many real applications. One example comes from genome-wide association studies. For a given phenotype, the associated genetic variants are usually a sparse subset of all possible variants. While the traditional method (e.g., Lasso) can find important individual genes, our method is able to find the sparse interaction between two (and potentially multiple) genes, which is especially desirable based on the biological knowledge that genes work together in structured groups [2].

[1.3. Novelty of our work] The key new realization of our paper is that, for our specific problem, it is possible to find the top 2k elements of the gradient without even calculating the entire gradient. This allows for our method, which is iterative hard thresholding (IHT) with approximate support recovery via count sketches, to run in sub-quadratic time.

The existing analyses of IHT does not carry over to our setting - because we have inexactness in finding the top-k AND inexactness in finding the gradient. Thus, our analysis has two main contributions: (1) showing that our count sketch based support recovery gives good approximation guarantees under the assumptions of restricted smoothness and sparsity of the optimal solution, and (2) showing that our IHT variant still converges linearly under this inexact support recovery. Thus our paper bridges the (so far separate) analyses of count sketch and stochastic iterative hard thresholding, for an important use case of finding higher order interactions.

**To Reviewer # 4** [2.1. Motivation.] The motivations and application are discussed in [1.1.] and [1.2.].

[2.2. Optimality of the current result.] $O(k \log p)$ samples, along with our landscape assumptions of restricted smoothness etc, are known information theoretic lower bounds for recovering a $k$-sparse $\Theta$. By Theorem 5, our method matches this lower bound up to a constant factor. Similarly, for the time and space complexity, the optimal complexity is $\Omega(kp)$, since a minimum of $\Omega(k)$ samples are required for recovery, and $\Omega(p)$ for reading all the entries. Corollary 6 shows that the time and space complexity of our method is $\widetilde{O}(k(k + p))$, which is near optimal. These results are briefly mentioned in the paper, we would like to highlight them in the revised version.

[2.3. More explanation for the statistical and computational trade-off. ] In our method, the parameter $b$ (which controls the output size of the count sketch) determines the statistical and computational trade-off. In Figure 1-(a), the black dashed line stands for solving the quadratic regression with exact gradient calculation, which is a statistical benchmark (not achievable in sub-quadratic time). By choosing larger $b$, we are getting closer to the that while paying extra computation (recall that the computational complexity is given by $\widetilde{O}(m(p + b))$, where $m$ is the batch size and $p$ is the dimension of input feature $\mathbf{x}$). As shown in Theorem 1 and Lemma 2, setting $b$ to the same order as sparsity $k$ is sufficient for the consistent parameter estimation. Setting $b$ to $p^2$ will yield exact gradient calculation while incurring quadratic complexity.

**To Reviewer # 5** [3.1. When the sparse assumption doesn't hold] Theoretically, the sparsity assumption is commonly adopted in high dimensional statistics. In the case when the ground truth is dense or the sparsity parameter $k$ is set to be smaller than the true sparsity $K$, some preliminary experiments indicate that both our method, and more classical ones like Lasso/standard IHT, can converge to a poor sparse solutions - unless there are some other extraneous assumptions. Thus lack of underlying sparsity in the true $\Theta^*$ is a problem for all sparse recovery methods.

Note that Corollary 6 shows that the overall time complexity is $\widetilde{O}(k(k + p))$, where $p$ is the dimension of input feature $\mathbf{x}$. A lower bound on the complexity is $\Omega(kp)$. Thus any method can have sub-quadratic complexity only when the sparsity $k$ of the truth is smaller than $O(p)$; this is both necessary and sufficient for our model as well.

[3.2. Motivations] The motivations and one real application where sparsity holds are discussed in [1.1.] and [1.2.].

# References

[1] James Jaccard and Robert Turrisi. *Interaction effects in multiple regression*, volume 72. Sage, 2003.

[2] Yun Li, George T O'Connor, Josée Dupuis, and Eric Kolaczyk. Modeling gene-covariate interactions in sparse regression with group structure for genome-wide association studies. *Statistical applications in genetics and molecular biology*, 14(3):265–277, 2015.


[Meta-Review · NeurIPS 2019]

This paper studies the quadratic regression problem, where the response is the sum of a linear and quadratic term. The naive approach to solving this on p-dimensional vectors would be to expand it to a regression problem on p^2 dimensional vectors. The main contribution of this paper is a new algorithm that achieves running time and space complexity that are subquadratic in p, building off of ideas about how to get subquadratic algorithms for the maximum inner product problem. The paper makes solid progress on a basic regression problem.